

# Detection of structural deficiencies in a global aerosol model to explain limits in parametric uncertainty reduction

Léa M. C. Prévost[1], Leighton A. Regayre[1,2,3], Jill S. Johnson[4], Doug McNeall[2], Sean Milton[1], Kenneth S. Carslaw[1]

[1]School of Earth and Environment, University of Leeds, Leeds, LS2 9JT, UK
[2]Met Office Hadley Centre, Exeter, Fitzroy Road, Exeter, Devon, EX1 3PB, UK
[3]Centre for Environmental Modelling and Computation, School of Earth and Environment, University of Leeds, Leeds, LS2 9JT, UK
[4]School of Mathematical and Physical Sciences, University of Sheffield, Sheffield, S3 7RH, UK

*Correspondence to*: Léa M. C. Prévost (eelp@leeds.ac.uk)

**Abstract.** Understanding and reducing uncertainty in model-based estimates of aerosol radiative forcing is crucial for improving climate projections. A key challenge is that differences between model output and observations can stem from uncertainties in input parameters (parametric uncertainty) or from deficiencies in model code and configuration (structural uncertainty), and these two causes are difficult to distinguish. Structural deficiencies limit efforts to reduce parametric uncertainty through observational constraint because they prevent models from being simultaneously consistent with multiple observations. However, no framework exists to detect structural deficiencies and assess their impact on parametric uncertainty. We propose a workflow to identify structural inconsistencies between observational constraints and diagnose potential structural deficiencies. Using a perturbed parameter ensemble, we sample uncertainty in aerosols, clouds, and radiation in the UK Earth System Model (UKESM), and evaluate model bias against in-situ observations of sulfate aerosol, sulfur dioxide, aerosol optical depth, and particle number concentration across Europe. Applying observational constraints reveals inconsistencies that no combination of the perturbed parameters can resolve. For example, sulfate concentrations in different regions cannot be matched simultaneously, and enforcing a compromise between region reduces skill across most variables. Additional examples include an inter-region inconsistency in $SO_2$ and an inter-variable inconsistency between aerosol optical depth and sulfate. By examining the parameter sets retained by constraints, we trace inconsistencies to the parameterisations that may cause them and propose targeted changes to address them. This approach offers a pathway for evidence-based model development that supports more robust uncertainty reduction and improves climate projection skill.

## 1. Introduction

Earth System Models are essential tools for understanding and projecting climate change. However, these models cannot directly resolve many complex or small-scale processes, such as cloud formation or aerosol-cloud interactions, due to computational restrictions. Instead, unresolved processes are represented using parameterisations: mathematical equations with adjustable input parameters that approximate physical behaviour. Different choices of parameter values lead to different model outputs, so the use of parameterisations inevitably introduces parametric uncertainty for quantities that cannot be observed



such as aerosol radiative forcing, which contributes to the spread in climate projections (Peace et al., 2020; Watson-Parris and
Smith, 2022).

Modelling centres often adjust parameter values to improve agreement with observations through *tuning*, which involves
expert-informed adjustments to a small number of key parameters to produce a single "best" parameter set for each model.
Tuning, however, relies on subjective decisions; modelling teams determine which simulated variables to prioritise, which
observations to use, and how to weigh them to best optimise their model (Hourdin et al., 2017). The result across multiple
models is a "collection of carefully configured best estimates" (Knutti et al., 2010) that reflect expert judgement and available
data, but not necessarily the full range of plausible outcomes. Although tuning is often necessary to produce stable and
physically realistic simulations (Schmidt et al., 2017), it obscures other causes of error in the model (Rostron et al., 2025).

These additional errors arise from the model's inherent structural limitations. All models depend on choices about which
physical processes to include, how they are formulated, the chosen spatial resolution used, and how the code is implemented.
Since no model is perfectly structured to represent the real world, all models carry some degree of structural uncertainty.
Structural uncertainty leads to model discrepancy or systematic error that cannot be resolved by adjusting parameters when
compared to observations (Goldstein and Rougier, 2004; McNeall et al., 2016; Sexton et al., 2012). As a result, there is a risk
that model tuning, when selecting parameter values that best match observations, will overcompensate for deficiencies in the
model's structure. The chosen parameter combinations may reproduce observations for the wrong reasons due to compensating
model errors. As a result, they will not produce reliable output when used under novel conditions, like when the model is used
for climate projections that inform policies (Golaz et al., 2013).

Understanding the causes of a model's structural uncertainty is an essential part of model development. However, it is
complicated by the fact that parametric and structural uncertainties are entangled, making it difficult to determine whether
discrepancies between model output and observations are due to parameter choices or deficiencies in the model's structure.
Historically, structural uncertainty has been explored using multi-model ensembles (MMEs, or model intercomparisons) by
comparing structurally different models (Collins, 2007; Flato *et al.*, 2013). However, each model in an MME is typically
subjectively tuned so only provides a limited view of its structure, as it is already pre-conditioned to match observations as
well as its structure allows. In addition, many models share common components or code, so the effective diversity within an
MME is often smaller than it appears (Masson and Knutti, 2011). The range of outputs generated by varying parameters within
a single model has been shown to be as large as, or even larger than, the spread across multiple models (Murphy *et al.*, 2004;



Yoshioka *et al.*, 2019), which suggests that MMEs alone provide only a partial picture of parametric and structural uncertainty,
and that a more systematic exploration of uncertainty is needed to separate these two main causes of model error.

The parametric uncertainty of a model can be sampled using a perturbed parameter ensemble (PPE). PPEs are created by
running the same model with different combinations of parameter values to capture the range of possible model outputs (Lee
*et al.*, 2011, 2012; Sexton *et al.*, 2012, 2021; Yoshioka *et al.*, 2019; Eidhammer *et al.*, 2024). The information derived from
PPEs can be extended using statistical emulators (e.g., Gaussian Process emulators) to predict model outputs for a much larger
set of parameter combinations than were simulated (O'Hagan, 2006). PPEs and emulators form a key part of the Uncertainty
Quantification (UQ) framework (Kennedy and O'Hagan, 2001), which aims to assess how different causes of uncertainty (e.g.,
parametric, structural, and observational) affect model output.

Within this framework, history matching is a method used to reduce parametric uncertainty. Rather than identifying a single
best-fitting parameter set, history matching rules out combinations of parameters that are observationally implausible, given
defined thresholds of the uncertainties in the quantities being compared (Craig et al., 1997). Unlike tuning, this method avoids
overfitting by retaining all parameter sets that remain observationally plausible. History matching has been applied both to full
climate models (Williamson et al., 2013) and to individual components such as the NEMO ocean model (Williamson et al.,
2017), land surface models (Raoult et al., 2024), as well as aerosol models (Johnson et al., 2020; Regayre et al., 2020).

History matching is designed to account for structural uncertainty. The "implausibility" of every model variant (a model run
with a different combination of parameter values) is calculated and used to determine which parameter combinations are ruled
out. The implausibility measure includes a structural error term as part of its definition. However, as there is no reliable way
to quantify structural uncertainty, this term effectively reflects the modeller's judgement about how wrong the model might
be (Williamson et al., 2015). If the term is too small, plausible parameter sets may be incorrectly ruled out; if it is too large,
implausible combinations may be retained. Consequently, the uncertainty in this term adds subjectivity to the process of ruling
out parameter combinations, without necessarily bringing us closer to disentangling parametric and structural uncertainty. As
a result, while history matching is more transparent than tuning because assumptions about uncertainty are explicitly stated, it
still carries limitations when structural uncertainty is poorly understood (Brynjarsdóttir and O'Hagan, 2014).

Unquantified structural uncertainties have limited the scientific community's ability to constrain uncertainty in predictions of
aerosol radiative forcing ($\Delta F_{aer}$), the change in Earth's radiative balance due to anthropogenic aerosol emissions. As the most



uncertain component of anthropogenic forcing (Forster et al., 2021), $\Delta F_{aer}$ complicates estimates of climate sensitivity to greenhouse gases and affects projections of global temperature change (Andreae et al., 2005), limiting how confidently we can simulate future climate change and inform policy decisions. Despite extensive use of observational constraints to reduce parametric uncertainty (Johnson et al., 2020; Regayre et al., 2023), uncertainty in $\Delta F_{aer}$ remains high (Regayre et al., 2025). Similar limitations have been reported in other recent studies, where applying large observational datasets led to only modest reductions in uncertainty in global-mean liquid water path adjustment (Mikkelsen et al., 2025) and effective radiative forcing from aerosol–cloud interactions, ($\Delta F_{aci}$, Song $et\ al.$, 2024), both of which contribute directly to the overall uncertainty in $\Delta F_{aer}$.

A clear illustration of the limits of observational constraints is found in Johnson et al. (2020), who used a history matching approach incorporating over 9,000 aerosol observations in an effort to substantially constrain $\Delta F_{aer}$. Yet, the resulting reductions in parametric uncertainty were minimal— 6 % for $\Delta F_{aci}$ (the component of $\Delta F_{aer}$ from aerosol–cloud interactions) and 34 % for $\Delta F_{ari}$ (the component from aerosol–radiation interactions). One reason for this limited constraint was that different observational datasets pulled model parameters towards opposite sides of their ranges, resulting in conflicting estimates of $\Delta F_{aer}$. These inconsistencies reduced the effectiveness of observational constraints, despite the size and diversity of the observational dataset, and suggest that we remain far from achieving the maximum feasible reduction in aerosol radiative forcing uncertainty.

Such inconsistencies are symptomatic of structural model deficiencies, as they indicate that the model cannot reproduce all available observations simultaneously. Evidence of similar inconsistencies was found in McNeall $et\ al.$, (2016), where constraining the climate model FAMOUS to match observations from the Amazon forest led to different parameter combinations being retained than when constraining the model to other forests. The model could represent features of individual forests, but its inability to represent all forests simultaneously implied that key processes are missing or overly simplified. The scale of this problem is systemic and substantial: in an attempt to reduce $\Delta F_{aci}$ uncertainty in the UK Earth System Model (UKESM1; Sellar $et\ al.$, 2019), Regayre $et\ al.$, (2023) found that only 13 out of 450 cloud and aerosol measurements could be used before structural inconsistencies started weakening the constraint, which indicates that some of the remaining parametric uncertainty might be due to unaddressed structural deficiencies. If such deficiencies were identified and addressed, more observations could be used and tighter bounds on $\Delta F_{aci}$ could potentially be achieved. Therefore, identifying the causes of inconsistent observational constraints and the structural deficiencies responsible for them is a necessary step towards improving model reliability and increasing model skill at simulating future climate.





There has been growing interest in using PPEs not only to quantify parametric uncertainty, but also to reveal structural
deficiencies that cannot be resolved by tuning parameter values alone (Carslaw et al., 2025). For example, Furtado *et al.*,
(2023) and Rostron *et al.*, (2023) used PPEs to explore parametric uncertainty in their models and detect discrepancies that
persist across all parameter combinations. Couvreux *et al.*, (2021) proposed a parameter calibration framework to identify
parameters which limit model performance by introducing structural uncertainty, to be implemented during model
development and tuning. Peatier et al., (2024) examined how variability across PPE simulations could provide information
about the presence of structural error. Despite these innovations, there is currently no agreed framework to identify structural
deficiencies that lead to conflicting observational constraints, and thus block progress in reducing parametric uncertainty.
Moreover, little attention has been given to identifying which model developments should be prioritised to most effectively
improve model skill at simulating future climate. Without such a framework, there is a risk that model developments increase
model complexity without delivering clear benefits (Proske et al., 2023).


In this study, we develop an approach to a) detect structural inconsistencies between observational constraints and b) identify
structural deficiencies that could cause them. We build on the work of Regayre *et al.*, (2023) who identified a key structural
inconsistency in observational constraints related to aerosol-cloud interactions. Our focus is on aerosol-radiation interactions
in European winter, where we explore the performance of a UKESM1 PPE by examining the effect of sulfate aerosol mass
concentration, sulfur dioxide concentration, aerosol optical depth, and particle number concentration as observational
constraints. Specifically, we aim to answer the following questions: 1) what are the main inconsistencies between these aerosol
observational constraints? 2) Can these inconsistencies help identify the structural deficiencies that limit our ability to reduce
uncertainty in $\Delta F_{\mathrm{aer}}$?

The paper is organised as follows: in Sect. 2 we outline our methodologies to identify inconsistencies and infer potential
structural deficiencies that may cause them. In Sect. 3.1 to 3.3, we evaluate the model's performance against in-situ
observations across the parametric space. In Sect. 3.4 to 3.6, we apply observational constraints and examine the
inconsistencies that arise. In Sect. 4, we identify priorities for structural model development and discuss how this approach
could be used more broadly to support uncertainty reduction in Earth system modelling.



## 2. Methods

We use the PPE and statistical emulation methodology described in Regayre *et al.* (2023). In Sect. 2.1, we summarise the components of the model configuration that are relevant to the study. Section 2.2 presents the measurements used to compute model bias. In Sect. 2.3, we outline how the main causes of parametric uncertainty were identified for each model grid box, and in Sect. 2.4, how this information informed the spatial clustering of the study region. Section 2.5 then details the calculation of model bias within each cluster, while Sect. 2.6 explains our approach to applying observational constraints. Finally, Sect. 2.7 defines the types and severities of observational inconsistency considered.

### 2.1. Experimental design

#### 2.1.1. Model version

The PPE used here was created using version 1 of the UKESM (UKESM1; Sellar *et al.*, 2019), which is based on the HadGEM3-GC3.1 physical climate model (Williams et al., 2018) and includes coupling to the United Kingdom Chemistry and Aerosol (UKCA) model (Archibald et al., 2020). Simulations were run using the atmosphere-only configuration, UKESM1-A, which consists of the GA7.1 atmosphere (Walters et al., 2019) with additional updates to aerosol, cloud, and atmospheric structure as described in Mulcahy *et al.*, (2020). The model resolution is N96 (1.875° × 1.25°, or approximately 208 km × 139 km at the Equator), with 85 vertical levels extending up to 85 km. Horizontal winds above approximately 2 km were nudged towards ERA-Interim reanalysis data for the period December 2016 to November 2017. Sea surface temperatures and sea ice were prescribed for the same period.

Each PPE member was forced using anthropogenic $SO_2$ emissions from the years 2014 and 1850, consistent with those used in CMIP6 (Eyring et al., 2016). Emissions of carbonaceous aerosol from residential and fossil fuel sources followed CMIP6 data for 1850, while present-day carbonaceous aerosol from biomass burning sources were prescribed using Copernicus Atmosphere Monitoring Service (CAMS) data for December 2016 to November 2017. Monthly mean output from a fully coupled UKESM simulation was used to prescribe ocean surface concentrations of dimethylsulfide (DMS) and chlorophyll, as well as atmospheric concentrations of gas-phase species, including OH and $O_3$. Volcanic $SO_2$ emissions included continuous and sporadic sources (Andres and Kasgnoc, 1998) and emissions from explosive eruptions (Halmer et al., 2002). Aerosol number concentrations were calculated prognostically using the GLOMAP-mode aerosol scheme (Mann et al., 2010, 2012), which represents five log-normal modes and includes sulfate, sea salt, black carbon, and organic carbon, internally mixed within each mode.



We use a version of UKESM1-A with structural changes described by Regayre *et al.* (2023). These include: a revised threshold
for ice mass fraction above which nucleation scavenging is deactivated to allow aerosol transport into the Arctic (Browse et
al., 2012); updated high-resolution lookup tables for aerosol optical properties (Bellouin et al., 2013), including mineral dust
(Balkanski et al., 2007) and improved aerosol absorption; and the inclusion of an organically mediated aerosol nucleation
parameterisation (Metzger et al., 2010), intended to improve the model's representation of remote marine and early industrial
aerosol conditions, known to affect the magnitude of $\Delta F_{\mathrm{aer}}$ (Carslaw et al., 2013).

### 2.1.2. Perturbed parameter ensemble and statistical emulation

The PPE from Regayre *et al.* (2023) consists of 221 model simulations, with 37 perturbed parameters related to aerosols,
clouds, and the physical atmosphere (detailed in Table A1). The selection of the perturbed parameters was based on those
identified in previous PPEs as large causes of uncertainty in key outputs (Regayre et al., 2015, 2018; Sexton et al., 2021;
Yoshioka et al., 2019), together with parameters associated with structural model developments (Mulcahy et al., 2018, 2020;
Walters et al., 2019). Their perturbation ranges were determined using formal expert elicitation using the Sheffield Elicitation
Framework (SHELF) approach described in Gosling (2018). The PPE was developed in two stages. In the first stage, the most
implausible parts of the parameter space were identified and removed by comparing simulated shortwave fluxes with
observations using a history-matching style approach. The second stage PPE was sampled from the remaining, more plausible
parameter space and forms the focus of this analysis.

Here, model output from the 221 PPE simulations, resolved at the grid-box level across Europe in January 2017, was used to
train statistical emulators for four variables related to aerosol–radiation interaction forcing: sulfate aerosol mass concentration
("sulfate"), sulfur dioxide concentration ($SO_2$), aerosol optical depth (AOD), and particle number concentration larger than 3
nm diameter ($N_3$). Gaussian Process emulators (O'Hagan, 2006) were constructed to represent the monthly mean of each
variable as a continuous function across the 37-dimensional input parameter space, with each parameter jointly varied over its
specified range (shown in Table A1). The emulators were then used to generate output for 1 million model variants at the grid-
box level, with a large reduction in computational cost compared to full climate model simulations. Emulator uncertainty was
quantified and assessed against the spread of emulator output (Fig. B1). Grid boxes where emulator predictive uncertainty
exceeded the spread in emulator output were excluded from the analyses to avoid relying on emulator predictions in regions
of high predictive uncertainty.



## 2.2. Measurements

We use in-situ aerosol measurements for January 2017 in Europe, aggregated to monthly means, for the four variables: sulfate, $SO_2$, AOD, and $N_3$. Measurements for sulfate, $SO_2$, and AOD were obtained from the Globally Harmonised Observations in Space and Time (GHOST) dataset (Bowdalo, 2024a; Bowdalo *et al.*, 2024b), which provides station-level monthly mean values. Sulfate measurements represent total particulate sulfate at the surface, reported in µg m⁻³. $SO_2$ concentrations were measured as surface-level sulfur dioxide in nmol mol⁻¹ and converted to µg m⁻³. AOD data are level 2.0 observations measured at a wavelength of 440 nm from the AERONET network (Sinyuk et al., 2020). $N_3$ represents the number concentration of particles larger than 3 nm, measured at the surface in particles per cm³. $N_3$ data were directly obtained from the European Monitoring and Evaluation Programme (EMEP, http: //ebas.nilu.no/, last access: 27 January 2025; (Tørseth et al., 2012)).

## 2.3. Causes of uncertainty

The importance of each parameter as a cause of model uncertainty was estimated using Generalised Additive Models (GAMs) at the grid-box level, following (Regayre et al., 2025). GAMs were fitted to emulated model output for each variable within individual grid boxes using the *pygam* Python package (Servén and Brummitt, 2018). The fitted GAM functions were used to quantify the variance in model output attributable to each parameter, while allowing for non-linear effects (Strong et al., 2014).

To quantify the parameter's contribution to output variance, we varied one parameter at a time across its sampled range while fixing all others at their median values. This approach isolates the marginal effect of the target parameter by removing variability introduced by changes in other parameters. The resulting 37 variances were summed to obtain the total parametric variance, and each parameter's contribution was expressed as a proportion of this total. The resulting percentage contribution to parametric uncertainty reflects both the range over which each parameter was perturbed and the local importance of that parameter to model output.

The GAMs were trained on the "unconstrained" subset of approximately 900,000 model variants, excluding those with *prim_so4_diam* values below ~10 nm, as defined in Regayre *et al.* (2025). In the original ensemble comprising 1,000,000 model variants, such low diameters led to implausibly high particle number concentrations, which were ruled out as observationally implausible by Regayre *et al.* (2023). Including these variants would have artificially inflated the apparent importance of *prim_so4_diam*, thereby masking the contributions of other parameters (Regayre et al., 2025).





### 2.4. Spatial clustering of causes of uncertainty

We applied k-means clustering, an unsupervised machine learning technique, to group grid boxes according to shared causes
of parametric uncertainty. The clustering was implemented using the *scikit-learn* Python package (Pedregosa et al., 2011), and
was based on the parameter percentage contributions to variance multiplied by the sign of variable dependence on parameter
values from the GAM fit (Sect. 2.3). The number of clusters was chosen iteratively: we began with a high number relative to
the size of the region (e.g. six for Europe) and reduced it if clusters showed redundant patterns in dominant parameters and
their contributions. In some instances, clusters that spanned wide regions remained undivided even as the number of clusters
increased. The clustering method preferentially split regions adjacent to grid boxes excluded for high emulator uncertainty
because of distinct local patterns in causes of uncertainty. In these cases, we manually divided large clusters by masking all
other grid boxes and applying k-means clustering again within the selected region following the same method.

### 2.5. Evaluation of model-observation bias within clusters

We evaluate model performance against observations within each cluster of shared causes of parametric uncertainty. For each
PPE simulation, we compute the mean model value over the set of grid boxes containing observations within the uncertainty
cluster, resulting in a cluster mean for each of the 221 PPE members. These cluster mean values are then used to train and
validate the emulator for each cluster (Fig. B2). Leave-one-out cross-validation indicates that the emulators reproduce cluster-
mean PPE outputs with high accuracy overall (e.g., NRMSE $\leq 0.09$), although some underprediction occurs for high values in
certain clusters (e.g., sulfate and $N_3$). These biases suggest that true values in these regions may be higher than emulated
estimates; however, given the focus on relative differences across clusters, these limitations are unlikely to affect the main
conclusions.

Model–observation bias is calculated for each model variant ($i = 1$ to 1,000,000) using normalised mean bias factors following
Yu *et al.*, (2006). Here, N is the number of observational sites in the cluster. Each site contributes a single observed value O,
collocated with one modelled value $M_i$ from each model variant. Thus, for a given model variant *i*, the cluster-mean model
value is $\overline{M_i} = \frac{1}{N}\sum_{j=1}^{N} M_{ij}$ and the cluster-mean observation is $\overline{O} = \frac{1}{N}\sum_{j=1}^{N} O_j$. The normalised mean bias factor ($B_{\text{NMBF}}$) is
calculated as follows:

$$B_{NMBF,i} = \begin{cases} 1 - \frac{\overline{O}}{\overline{M_i}}, & \text{if } \overline{M_i} < \overline{O} \\ \frac{\overline{M_i}}{\overline{O}} - 1, & \text{if } \overline{M_i} > \overline{O} \end{cases} \qquad (1)$$



### 2.6. Application of observational constraints

The steps in Sect. 2.5 provide the model–observation bias for each of the 1,000,000 model variants. Observational constraints are then applied by retaining only those variants with the smallest absolute $B_{\mathrm{NMBF}}$, which correspond to those closest to the mean observed value. We apply observational constraints to the original set of 1,000,000 model variants, rather than the "unconstrained" subset of ~900,000 used for clustering (Sect. 2.4). While low *prim_so4_diam* values are excluded from uncertainty analyses due to their unrealistic nature, including them here helps illustrate the effect structural deficiencies in

observational constraints.

Observational uncertainties are not directly incorporated into the constraint process. Instead, we retain a threshold of 5,000 model variants (0.5 %) closest to observations to prevent over-constraint, given the presence of unquantified measurement errors. This threshold was also used by Regayre *et al.* (2023), and was chosen to approximate the proportion of model variants

retained using a more rigorous history matching approach that explicitly accounts for observational uncertainty, emulation uncertainty and other model-to-observation comparison uncertainties (Johnson et al., 2020; Regayre et al., 2020). In this research, observational constraints are not used to identify a single "best" model variant or to quantify parametric uncertainty. Rather, they are used as tools to explore model responses to constraints and to identify potential structural deficiencies.

For joint observational constraints, we identify the model variants that are common to each individual constraint set. In cases where no common variants are found, we define the constraints as inconsistent, using definitions that follow in Sect. 2.7. To explore the extent of the inconsistency and assess how conflicting constraints might be accommodated, we progressively relax individual constraints until at least around 300 model variants are retained in the overlapping set. We define this as a compromise between inconsistent observational constraints, following Regayre *et al.* (2023).


When observations are outside of the range of the model output of PPE members, they are not used in the calculation of model-observation bias (Sect. 2.5) and are therefore not included in the process of observational constraints. An observation outside the PPE range is a clear indication of the presence of a structural model deficiency, as it means that no amount of parameter retuning will bring the model into agreement with the observations. In these cases, we provide hypotheses on potential

consequences for our results.

While observations outside the PPE range are excluded from the constraint process, they are retained for evaluation purposes. Because these values lie beyond the range represented by the ensemble, they cannot be meaningfully used for constraint.



However, they remain important for assessing model skill and identifying potential structural limitations. To ensure a complete evaluation, we assess the impact of each observational constraint on model–observation bias across all available observations, including those outside the PPE range. For example, when constraining using $SO_2$, only $SO_2$ observations within the PPE range for all regions are used in the constraint, but model skill is evaluated using all available observations for sulfate, AOD, and particle number concentration, even those outside the PPE range. Similarly, when constraining toward AOD, we use only AOD observations within the PPE range for the constraint, but evaluate model skill against all sulfate, $SO_2$, and particle number observations.

### 2.7. Definitions of potential structural inconsistencies

In the ideal case, all observational constraints would guide the model toward the same part of parameter space. That is, each constraint would support convergence toward the parameter combination that best represents the real system. When constraints do not converge, it indicates that the model would need to be tuned differently to match each variable and that, having exhausted the parameter space, no model exists that is consistent with multiple observations. In history-matching terminology, this is referred to as the "terminal case" (Salter et al., 2019). In such cases, the model is not realistic which suggests a potential structural deficiency. We define this lack of convergence between constraints as a *structural inconsistency*.

The concept is related to Keith Beven's definition of a *behavioural model*, where a parameter set is considered "behavioural" if it cannot be rejected as observationally implausible (Beven, 2006). In our context, we identify cases where the model may be *partially behavioural* (i.e., satisfying individual constraints) but not *universally behavioural* across different aspects of the model (e.g., variables, regions).

Here, we define two *levels* of structural inconsistency to characterise ways in which convergence may fail (Fig. 1).

- Level 1 inconsistencies happen when observational constraint of one aspect of the model degrades performance in another, and vice versa – i.e., making the model skilful for one aspect makes it less skilful for another. In this case, although the two constraints do not converge, there exist model variants (parts of parameter space) capable of matching both observations simultaneously, but these variants are on average less skilful for both aspects than for either when considered individually.

- Level 2 inconsistencies happen when the constraint of one aspect of the model eliminates any agreement with another. In this case, there exist no model variants capable of matching both observations simultaneously, meaning that no combination of parameters can satisfy both constraints at once.



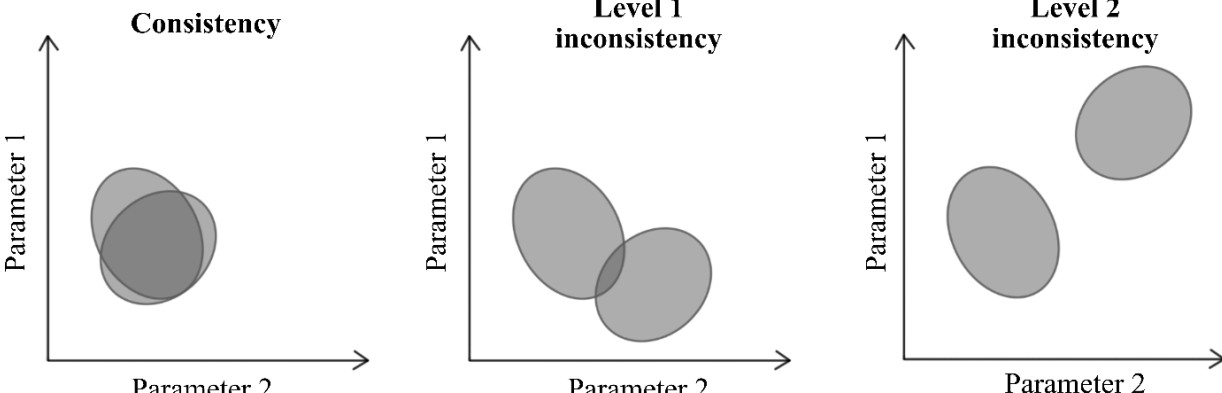

**Figure 1. Schematic showing the possible levels of inconsistency between two observational constraints.** The shaded regions are the parts of parameter space that match one observation type. The diagram only represents the 2-dimensional aspects of what is in our case a 37-dimensional problem.

We also distinguish *types* of inconsistency: inter-variable (between different observed variables) and inter-regional (the same variable observed in different regions, defined as clusters of grid boxes that share dominant causes of uncertainty).

We interpret the existence of an inconsistency as evidence of a potential structural deficiency in the model. However, such an inconsistency is not definitive proof of structural error; other explanations are possible, including larger than estimated observational error or the possibility that important parameters have not been perturbed. Conversely, not finding an inconsistency does not guarantee that the model is free from structural deficiencies. Some errors may only be detectable under specific model setups, such as with different spatial or temporal resolutions, or when perturbing different parameters. Our approach allows us to identify and address those inconsistencies that are detectable, and exploring plausible reasons for them provides actionable information for guiding model development priorities.

## 3. Results

We divide the analysis into six steps to identify potential structural inconsistencies in the model and assess their impact on model skill. First, we assess the parametric uncertainty ranges of the PPE for sulfate, $SO_2$, AOD, and $N_3$ in January, and compare them to observations to provide a baseline for understanding model behaviour and bias (Sect. 3.1). Second, we identify the key parameters driving uncertainty by clustering model grid boxes over Europe into sub-regions based on shared causes of uncertainty (Sect. 3.2). Third, we quantify model-observation biases within each uncertainty cluster before applying




observational constraints (Sect. 3.3). We then provide an example of using sulfate concentration observational constraints to
reveal a potential structural inconsistency between two regional clusters (Sect. 3.4). We explore the consequences of this
inconsistency when combining constraints (Sect. 3.5). Finally, we extend the analysis to identify other structural
inconsistencies across the variables and discuss their implications for model skill (Sect. 3.6).

### 3.1. The model, its parametric uncertainty and comparison with observations

Prior to emulation, we begin by quantifying the average magnitude of model variables and their variability across the PPE in
Fig. 2, which shows the PPE median (left column) and inter-quartile range (right column). Average sulfate and $SO_2$
concentrations are highest in the Balkans and Eastern Europe, near anthropogenic emission sources. In that region, sulfate
concentrations range from 7 to 20 µg/m³, and $SO_2$ concentrations range from 20 to 100 µg/m³. Particle number concentration
is also highest across mainland Europe, with median values from 7,000 to above 25,000 cm$^{-3}$ in Eastern Europe and between
3,000 and 10,000 cm$^{-3}$ in Western Europe. For AOD, the highest median values are near volcanic emission sources in Southern
Italy (between 0.5 and 0.6) and near sea salt emission sources over the North Sea and the Atlantic Ocean (around 0.3).

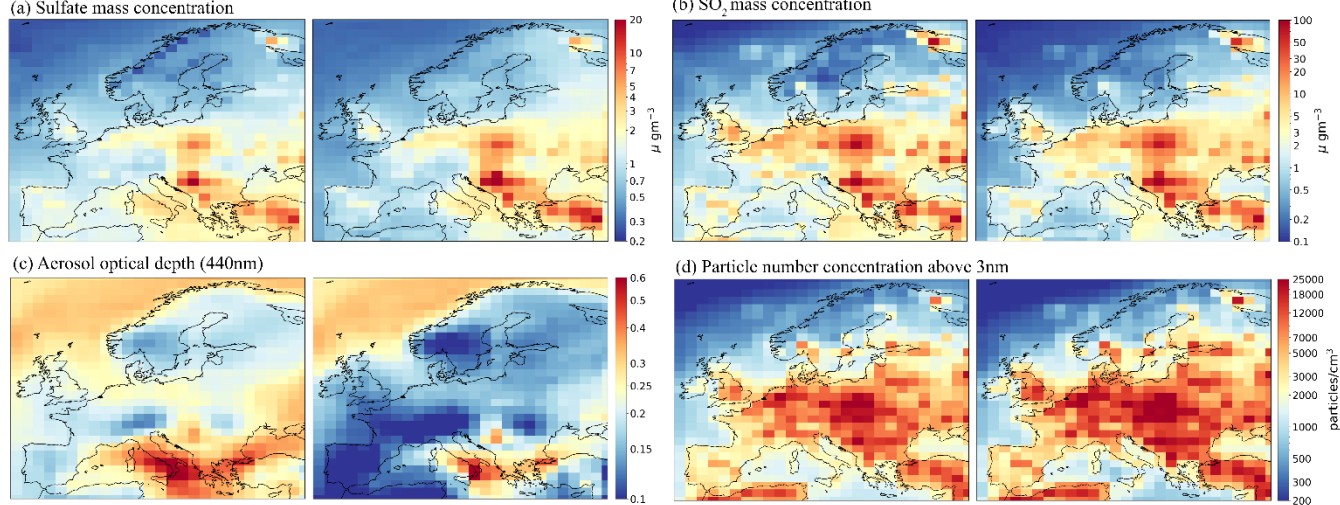

**Figure 2. PPE median (left) and interquartile range (right) for the four model variables** in January 2017 across Europe.

The interquartile ranges for sulfate, $SO_2$, and particle number concentrations follow the same spatial pattern as the median,
with higher uncertainty in regions where the median is also high. However, the interquartile range for AOD has a lower value
than its median inland (IQR = 0.1 but median = 0.15), except in Southern Italy and Greece (Fig. 2c). This difference may be
because AOD integrates contributions from multiple aerosol types, but only a subset was perturbed in the PPE (e.g., sea salt



and sulfate, but not dust, nor carbonaceous aerosol), which may have limited the variation across ensemble members relative to the median.

We next assess how well the model perturbed parameter range overlaps with in-situ observations for each model variable. Figure 3 shows in-situ observations relative to the empirical distribution of the PPE output across the 221 members.

Observed sulfate concentrations are well represented by the model across the perturbed parameter space. In Fig. 3a, most sulfate concentration observations are within the 90 % credible interval of the PPE distribution. One exception is a site in Slovakia, where observed concentrations are lower than all modelled values in a region with relatively high sulfate (Fig. 2a).

Overall, the model parameter uncertainty spans sulfate concentrations at each station.

For $N_3$ (Fig. 3d), most observations are within the PPE range; however, three observations from Southern France, Switzerland, and Northern Italy are below the PPE distribution, indicating that all PPE members overestimate particle number concentration at these sites. In addition, two nearby observations are positioned near the lower edge of the PPE range (between the 5th percentile and the distribution boundary), which suggests that modelled $N_3$ is consistently overestimated in this region.



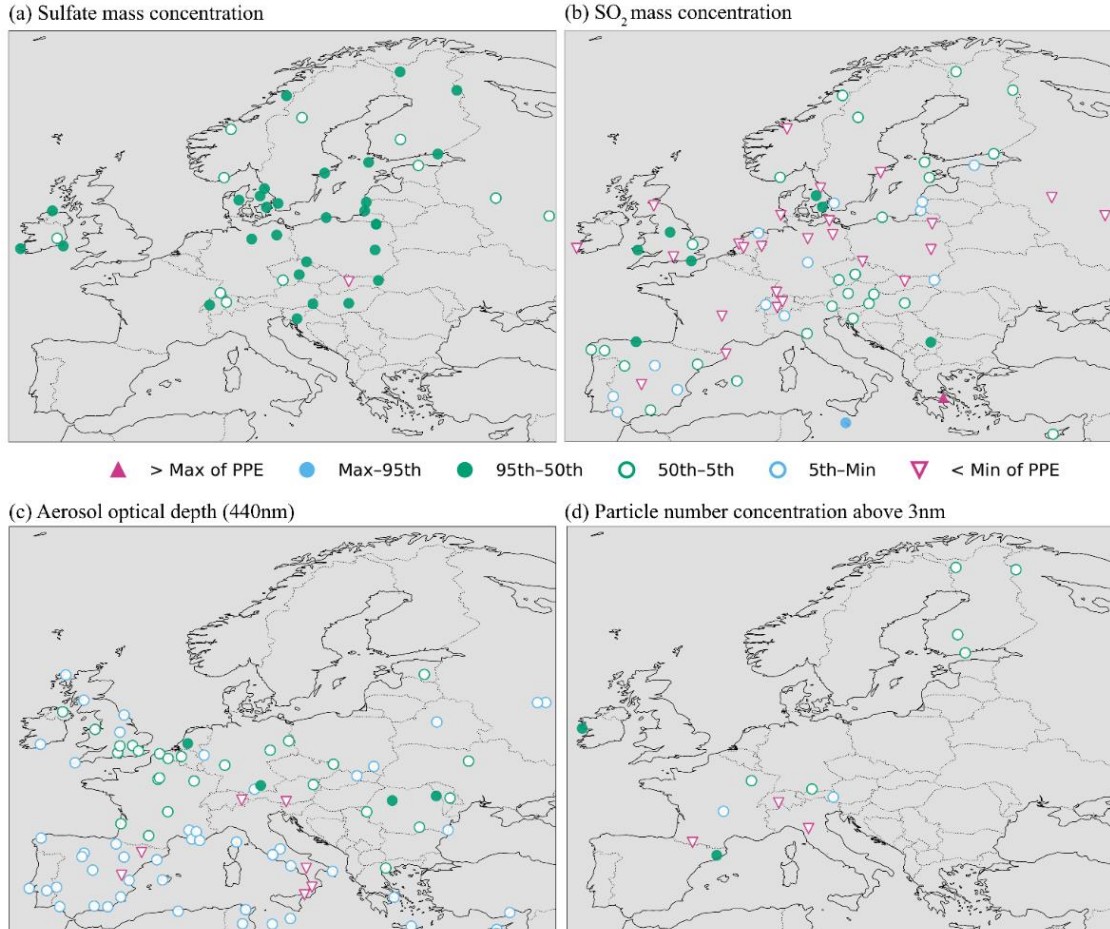

**Figure 3. Observed values and their position within the PPE range in January 2017 across Europe for the four variables.** Triangles indicate observations outside the PPE range. Circles represent observations within the PPE range.

Figure 3b shows that the model overestimates $SO_2$ concentrations at most measurement stations, with many observations below the lowest PPE member. This PPE-observation discrepancy suggests that the model has a structural deficiency that causes a high $SO_2$ concentration bias over central Europe, that cannot be overcome by perturbing the parameters in this PPE. Outside this area, some observations are within or near the 90 % credible interval. A plausible source of this structural deficiency is the emission height treatment in UKESM1, where all anthropogenic $SO_2$ emissions are injected at the surface rather than distributed vertically. This treatment leads to higher anthropogenic $SO_2$ concentrations close to source regions, but that are more efficiently removed by dry deposition (Mulcahy et al., 2020).



Modelled AOD is also mostly overestimated, particularly in Southern Europe. Figure 3c shows that observations in Southern Italy, in Spain and in the Alps are lower than the PPE range. In addition, all observations around the Mediterranean and in Spain are below the 5th percentile of the PPE distribution. The largest bias in Southern Italy is near the Mount Etna volcano, which is continuously degassing and sporadically erupts. This suggests the reason for the bias is likely to be related to choices
made during model configuration: continuous volcanic emissions were prescribed as an average over 1970s-1997 (Andres and Kasgnoc, 1998), but were compared here to observations from January 2017, when volcanic activity near Mount Etna was lower than average (Delle Donne et al., 2019). This PPE-observation discrepancy may also affect comparisons over the wider Mediterranean region where observations are close to the lower edge of the PPE distribution. Some observations in the UK, central and Eastern Europe are also below the 5th percentile of the PPE distribution, which suggests that the model also
overestimates AOD overall in Europe.

Structural deficiencies manifesting as observations outside the parameter uncertainty range are the easiest to detect. In the rest of the paper, our goal is to move beyond these most obvious cases and identify more subtle indicators of potential structural deficiency.

**3.2. Clusters of shared causes of parametric uncertainty**

In this section, we explore the regional causes of parametric uncertainty by grouping grid boxes into clusters that share causes of parametric uncertainty (Sect. 2.4). In Fig. 4 and 5, every grid box within a cluster is influenced by the same set of key parameters, with approximately the same contribution from each parameter. Therefore, we expect an observational constraint within a cluster to reduce uncertainty across the cluster. However, this effect is not guaranteed to be uniform: while a parameter
may contribute a similar amount of uncertainty in different grid boxes, the model's sensitivity to that parameter (the local gradient) can vary with local conditions, which could lead to differences in the degree of uncertainty reduction within the cluster.

Our methodology allow us to identify (1) which parameters contribute most to model uncertainty in the set of ~900,000 model
variants (Sect. 2.3) in each region, and (2) define sub-regions that can be compared against one another to investigate inter-region inconsistencies (Sect. 2.4). We chose to compare clusters instead of geographic boundaries because geographic regions are arbitrary and may group grid boxes influenced by very different aerosol processes. Clustering based on shared causes of uncertainty is a better reflection of the model's underlying processes and should therefore help identify spatial structural



inconsistencies. Here, we describe the most important causes of parametric uncertainty for sulfate concentrations, SO$_2$
concentrations, AOD, and N$_3$ across the uncertainty clusters.

(a) Sulfate mass concentration

(b) SO$_2$ mass concentration



**Figure 4. Clusters of shared causes of parametric uncertainty for (a) sulfate concentration and (b) SO₂ concentrations in January 2017.** Based on the sample of around 900,000 model variants (parameter combinations) after removing *prim_so4_diam* < 10 nm. The legend identifies the first 4 key parameters driving uncertainty in each cluster. The percentage of variance caused by each parameter is shown in the pie charts, displayed anticlockwise from most to least important. Masked grid boxes (in white) indicate regions where emulator uncertainty exceeds the spread of the emulated response, see Fig. B1.


Figure 4a shows that more than 75 % of parametric uncertainty in sulfate concentration in Europe is caused by the dry deposition of the accumulation mode aerosol (*dry_dep_acc*, a loss process) and the acidity of cloud droplets (*cloud_drop_acidity*, a parameter that affects the O₃ + SO₂ → sulfate oxidation rate in cloud water (Turnock *et al.* 2019)) in

each cluster. However, the order of importance and proportions of sulfate uncertainty caused by these parameters changes between clusters, as do contributions from other key parameters. In Central Europe (green cluster), the region with the highest sulfate emissions (Fig. 2a), *cloud_drop_acidity* contributes most to parametric uncertainty (60 %), likely because the region is inland and polluted, hence cloud acidity has a stronger influence on sulfate formation. In Northern Europe (blue cluster), *dry_dep_acc* contributes more to parametric uncertainty (65 %) than *cloud_drop_acidity* (25 %), likely due to the remoteness of the region which allows sinks to have a larger influence on concentrations than sources. There is also a small contribution

from *dms* as a source (3 %), likely due to the proximity of this region to the Atlantic Ocean. In Western Europe (pink cluster), *dry_dep_acc* and *cloud_drop_acidity* contribute equally to uncertainty (around 40 %). The Mediterranean region (red cluster) is distinct as partly influenced by SO₂ emissions from volcanic sources (*volc_so2*, 10 %) and the yellow cluster appears to surround the white grid boxes excluded due to high emulator uncertainty.


Figure 4b shows that SO₂ concentration over Europe is mainly controlled by the regional anthropogenic SO₂ emission rate parameter (*anth_so2_eur*) and parameters that affect its atmospheric lifetime by deposition (*dry_dep_so2*) or loss by formation of sulfate (*cloud_drop_acidity*). In Central Europe (green cluster), the most important contributors are *dry_dep_so2* (65 %) and *anth_so2_eur* (18 %), which may reflect local anthropogenic emissions that likely drive the high SO₂ concentrations seen

in Fig. 2b. In Western Europe (pink cluster), *anth_so2_eur* is most important (46 %), consistent with moderately high anthropogenic emissions from the UK and Spain. In Scandinavia (dark blue cluster), *anth_so2_eur* is slightly less important (13 %), ranked third after *dry_dep_so2* (50 %) and *cloud_drop_acidity* (30 %), which could reflect the more remote nature of the region. These same parameters drive uncertainty, in different combinations with other key parameters, in the grid boxes surrounding Spain (orange cluster), Eastern Europe (yellow cluster), and the marine (light blue) cluster.






**Figure 5. Clusters of shared causes of parametric uncertainty for (a) aerosol optical depth and (b) particle number concentration in January 2017.** All other features are identical to Fig. 4.

Figure 5a shows the main parameters causing uncertainty in AOD in Europe are *sea_salt* (natural source), *dry_dep_acc* (deposition), and *cloud_drop_acidity* (formation of sulfate aerosol). In Central Europe (green cluster), *cloud_drop_acidity*,





*sea_salt*, and *dry_dep_acc* contribute similar amounts (more than 20 % each), suggesting that sulfate aerosol formation and deposition processes contribute to AOD along with natural sources. In the Atlantic Ocean and Northern Europe (blue cluster), *sea_salt* contributes most to parametric uncertainty (75 %), likely due to strong marine influence and winds transporting sea salt particles inland. Clusters around the Mediterranean (purple and orange) are both influenced by *volc_so2* (around 9 %),

which suggests that the PPE-to-observation discrepancy linked to volcanology shown in Fig. 3c is likely to extend to all observations in the red and purple clusters (all of which are below the 5th percentile of PPE values).

Figure 5b shows the main parameters causing uncertainty in $N_3$ in Europe are *carb_ff_diam* (diameter of carbonaceous aerosol from fossil fuels), *prim_so4_diam* (diameter of sub-grid-scale sulfate particles at emission), and *dry_dep_acc* (deposition).

Parameter controlling the size of particles are most important near point emission sources: for a fixed emission mass flux, reducing the size of particles will increase the number of aerosol particles emitted. In Central Europe (green cluster), *carb_ff_diam* and *prim_so4_diam* and contribute most to parametric uncertainty (each around 20 %), suggesting that source emissions dominate in polluted regions. In the Atlantic Ocean and Northern Europe (blue cluster), *dry_dep_acc* contributes most (36 %), likely due to the region being more remote, and thus having a higher proportion of accumulation mode aerosol.

In Western Europe and the Mediterranean region (pink cluster), contributions are similar to the other two clusters, with a small additional contribution from the offline oxidant OH concentration scaling factor (*oxidants_oh*, 7 %).

The uncertainty clusters described in this section will be used as sub-regions of Europe throughout the rest of the paper to evaluate whether observational constraints are consistent across clusters. Clusters with too few observations (sulfate: red, yellow; $SO_2$: light blue, orange; $N_3$: pink), or where all observations are outside the range of the PPE ($SO_2$: yellow), are

excluded from further analysis.

### 3.3.  Model-observation bias in uncertainty clusters

In this section, we evaluate model–observation bias using the normalised mean bias factor ($B_{NMBF}$, Eq. (1)) across the parametric range within each cluster of shared causes of uncertainty identified in Sect. 3.2. Evaluating $B_{NMBF}$ for the original

distribution of model variants helps assessing model skill across clusters and variables and forms the basis for applying observational constraints and detecting structural inconsistencies.





Figure 6 shows boxplots indicating the distribution of $B_{NMBF}$ across the 1,000,000 emulated model variants for each variable, within their respective uncertainty clusters: pink (Western Europe), green (Central Europe), blue (Northern Europe), and

orange (Southern Europe). Although clusters do not map exactly onto the same region for each variable, we refer to them in this way for ease of comparison. The largest biases are in $N_3$, AOD and $SO_2$ which are all biased high in the model on average. For sulfate, there is more regional variation with some clusters biased high and others biased low. While we highlight median biases (the horizontal line inside the box, and Table 1) to capture general tendencies, the full distributions of model variants span observed values, which suggests that consistent observational constraint across variables and regions remains feasible at

this stage.

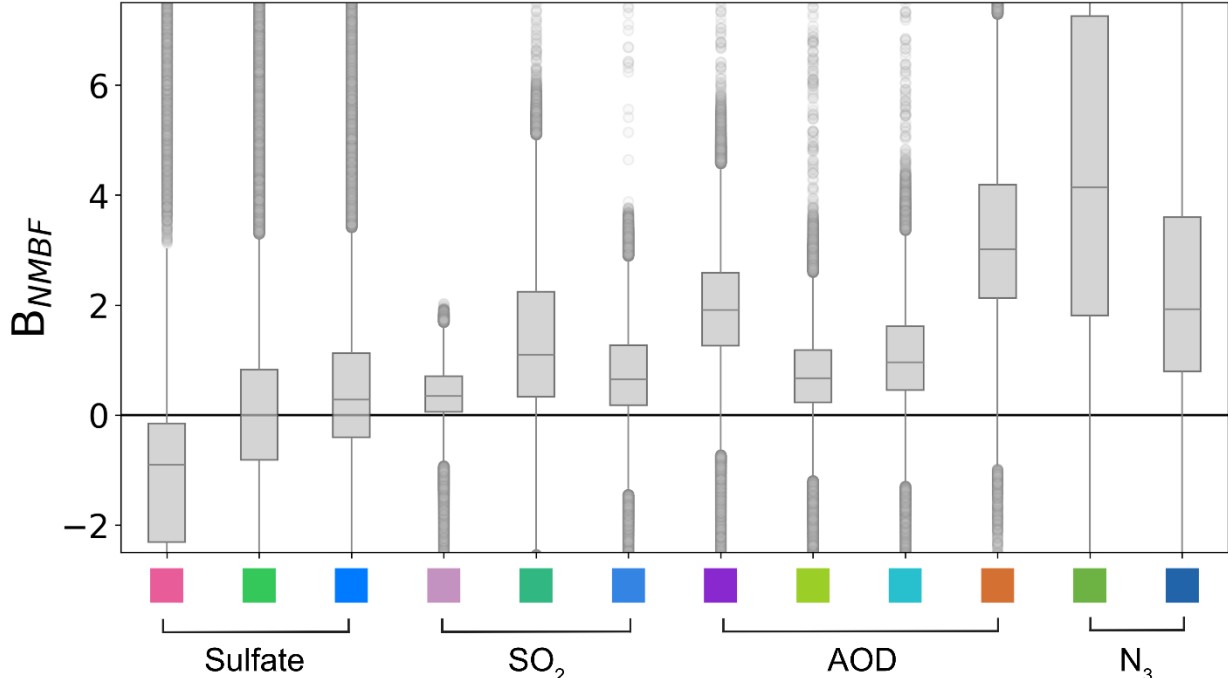

**Figure 6. Boxplots to show the distribution of the normalised model-observation mean bias factor ($B_{NMBF}$) over the original set of model variants, across variables and clusters.** The coloured patches on the x-axis correspond to the cluster colours from Fig. 4 and 5. The boxes show the interquartile range (IQR) of the distribution. The horizontal line inside the box shows the median (50th percentile). The

whisker (vertical line) extends to $1.5 \times$ IQR. The data points are the model variants outside the whisker's range (outliers). Cluster colours are consistent across regions: green for Central Europe, blue for Northern Europe/Scandinavia, and pink for Western Europe (UK/Spain).

**Table 1. Median normalised model-observation mean bias factor ($B_{NMBF}$) over the original set of model variants, across variables and clusters.** Where two values are given, the second in brackets corresponds to the median bias after excluding observations outside the PPE range. Clusters with too few observations to evaluate are indicated with a dash (–).





| Variable | Pink Cluster, Western Europe | Green Cluster, Central Europe | Blue Cluster, Northern Europe | Orange Cluster, Southern Europe |
|---|---|---|---|---|
| Sulfate | -0.90 | -0.01 (-0.24) | 0.28 | – |
| $SO_2$ | 0.35 (-0.02) | 1.10 (0.80) | 0.65 (0.43) | – |
| AOD | 1.91 (1.73) | 0.67 (0.60) | 0.96 | 3.01 (2.45) |
| $N_3$ | – | 4.14 (2.66) | 1.92 | – |


$N_3$ is highly overestimated in Central and Northern Europe. On average, particle number concentration in Central Europe has a higher positive bias (green cluster, bias = 4.14) than in Northern Europe (blue cluster, bias = 1.92). After excluding observations outside the PPE range in the green cluster, the bias decreases to 2.62. Boundary layer nucleation was included in this model version (with perturbed rates) using the organically mediated scheme of Metzger et al. (2010), which is not

implemented in the release version of UKESM1. The lack of new particle formation in the release version likely result in lower apparent bias than reported here. However, this result indicates a structural deficiency in the representation of particle number.

AOD is overestimated across all clusters. On average, the bias is highest in Southern Europe (orange cluster, bias = 3.01) and Western Europe (pink cluster, bias = 1.9), and smaller in Northern Europe (blue cluster, bias = 0.96) and Central Europe (green

cluster, bias = 0.67). After excluding observations outside of the PPE range that are associated with clear structural deficiencies, likely related to volcanic emissions (Fig. 3 and Sect. 2.6), average model bias decreases in all clusters, but AOD remains overestimated. The region-wide positive bias suggests that the model is systematically overestimating aerosol sources, size or radiative properties. Possible explanations include: (a) carbonaceous aerosol emissions being too high across Europe (we perturbed emission diameters but not emission mass fluxes); (b) inaccuracies in aerosol radiative properties, potentially tied to

incorrect size distributions; or (c) sea salt emissions being overestimated—especially since sea salt dominates parametric uncertainty in the blue and red clusters (Fig. 5a).

Surface $SO_2$ concentrations are generally overestimated by the PPE across Europe, although the magnitude of the overestimation varies by region. On average, $SO_2$ is overestimated in Western Europe (pink cluster, bias = 0.35), approximately

twice as much in Northern Europe (blue cluster, bias = 0.65), and again nearly double in Central Europe (green cluster, bias = 1.10). As shown in Fig. 3b, many $SO_2$ observations, are outside the PPE range, particularly in Central Europe, which drives the large overestimation in all clusters. We hypothesise that this bias arises from the model's treatment of anthropogenic $SO_2$ emissions, which are injected at the surface rather than vertically distributed (Sect. 3.1). After excluding observations outside



of the PPE range, model bias decreases. On average, the bias in the pink cluster approaches zero (bias = -0.02), while the blue
and green cluster remain overestimated, following a similar ratio: green (bias = 0.80) is approximately double that of blue (bias
= 0.43).

For sulfate concentrations, the sign and magnitude of model bias are region-specific. On average, sulfate is underestimated in
Western Europe (pink cluster, bias = -0.50) and to a lesser degree in Central Europe (green cluster, bias = -0.13), yet is
overestimated in Northern Europe (blue cluster, bias = 0.18). The fact that concentrations are overestimated in some regions
and underestimated in others, may point to missing emission sources or to regionally varying production and removal processes
that are not fully captured by the model, which could suggest a need for regime-aware parametrisations (e.g. Qian *et al.*, 2024).
However, there may be parts of the sampled parameter space that minimise the biases in all three clusters, which we explore
in Sect. 3.4.

**3.4.  Inconsistency between observational constraints**

We now assess inter-region consistency when applying observational constraints (Sect. 2.7). First, we present our
categorisation of inter-region inconsistencies (between uncertainty clusters) using model constraint to observed sulfate
concentrations as an example. Then, we extend our analysis to evaluate inter-region inconsistencies for other variables.

Figure 7 shows how constraining the model to sulfate mass concentration observations in one region affects model skill at
simulating sulfate concentrations elsewhere. Constraint to match observations in one region achieves near-perfect agreement
there at the expense of degrading model skill elsewhere in all cases. In the original distribution, sulfate concentrations in
Western Europe (pink cluster) are underestimated on average across the parameter space. When the model is constrained to
match observations in the pink cluster, sulfate concentrations increase not only in that cluster, but also in the green and blue
clusters, which increases their existing positive biases (Fig. 7a). The opposite happens when the model is constrained to the
Northern Europe (blue) cluster, where concentrations are on average overestimated: sulfate concentrations decrease across all
regions, including the pink and green clusters, again increasing the negative bias in those clusters (Fig. 7c). The model's only
response to regional constraints is to shift sulfate concentrations across the continent, which means that adjusting
concentrations in one region inevitably affects others.





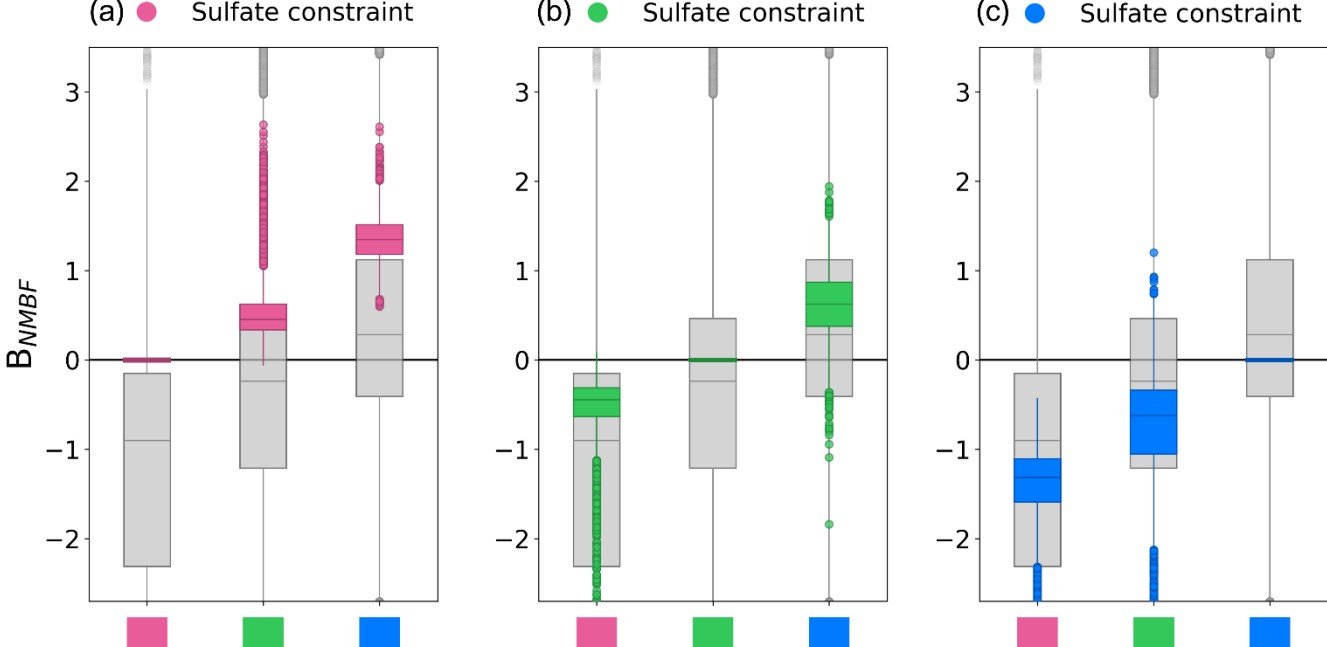


**Figure 7. Boxplots to show the effect of observational constraint on the distribution of the normalised model-observation mean bias factor ($B_{NMBF}$) in sulfate concentration after observational constraint.** The grey boxplots are identical to those in Fig. 6, for sulfate clusters in Fig. 4a. Overlaid coloured boxplots show the $B_{NMBF}$ distribution for the 5,000 model variants closest to observations for sulfate mass concentration in (a) the pink cluster (Western Europe) and (b) the green cluster (Central Europe) and (c) the blue cluster (Northern

Europe).

The opposing effects of these regional observational constraints clearly illustrate the model's inability to represent regional variations in sulfate concentrations simultaneously, even though the model simulations sample combinations of 37 parameters. Parameter combinations that improve agreement in one region entirely remove agreement in another: after applying observational constraints from either pink or blue clusters, the $B_{NMBF}$ distribution for the other no longer crosses zero. As a

result, the model can either reproduce observed sulfate concentrations in Western Europe or in Northern Europe, but not both simultaneously, when using a global-mean approach to aerosol processes (which is the case with aerosol removal parameter, *dry_dep_acc*, and the cloud droplet acidity parameter, *cloud_drop_acidity*). The inability to identify parameter sets that simultaneously satisfy constraints across clusters is evidence of a level 2 inter-region structural inconsistency in sulfate concentrations (Sect. 2.7).





We further analyse the effect of observational constraints on other variables and clusters using two metrics: percentile position of the observation within the model distribution and median $B_{NMBF}$, as exemplified in Fig. 8. The change in median $B_{NMBF}$ shows whether the centre of the distribution of model variants shifts towards or away from the observed value after applying the constraint. However, a lower absolute median $B_{NMBF}$ could be achieved by increasing precision without increasing

accuracy, so even though the average bias is reduced, the distribution may not span the observed value (e.g. blue constraint in Fig. 8 example schematic). Thus, we additionally use the percentile position of the observed values to simultaneously quantify the effect of constraints on precision and accuracy.

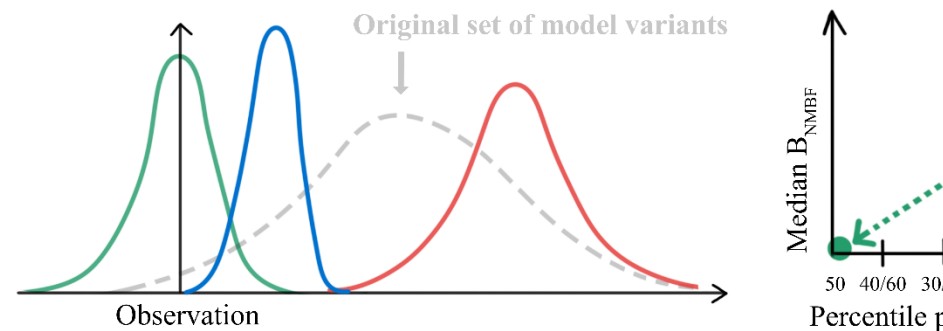
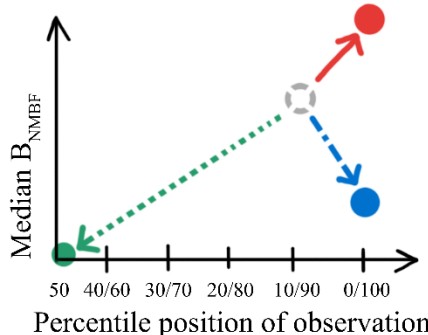

**Figure 8. Schematic illustrating the two metrics used to assess the effect of observational constraints on other variables and clusters:**
**the median $B_{NMBF}$ (distribution shift) and the percentile position of the observation within the model distribution.** Three example observational constraints are applied to the original set of model variants in dashed grey. The green distribution shows a case where both metrics improve; the red shows both worsening; and the blue shows an improvement in median $B_{NMBF}$ (higher precision) but a shift in the observation's percentile position away from the distribution centre (lower accuracy).

Figure 9 shows the effect of sulfate constraints presented in Fig. 7 on both precision and accuracy. In Fig. 9a, constraining the
model to the observations in the pink cluster selects the variants closest to that observation. As a result, the pink arrow points to the origin, indicating near-zero mean bias and a percentile position of the observation near the centre of the constrained model distribution. However, the effect of this constraint on the green and blue clusters in Fig. 9a is to increase median $B_{NMBF}$ (shown by arrows pointing upwards) and to shift the observation's percentile position away from the distribution centre (arrows pointing to the right). The green and blue arrows point to 0/100, meaning that the observed sulfate concentration is outside the
constrained distribution. The same pattern is seen in Fig. 9c when constraining to the blue cluster: Although the arrows point downward, they still move away from the zero line, indicating a larger (negative) median bias and reduced agreement. After applying the green constraint (Fig. 9b), the median $B_{NMBF}$ in the pink cluster improves slightly, but the observation is outside the constrained distribution, meaning that no model variants match observations in that region. In the blue cluster, both the



median $B_{NMBF}$ and the percentile position of the observation worsen. So, improving agreement in one cluster worsens
agreement in the other two.

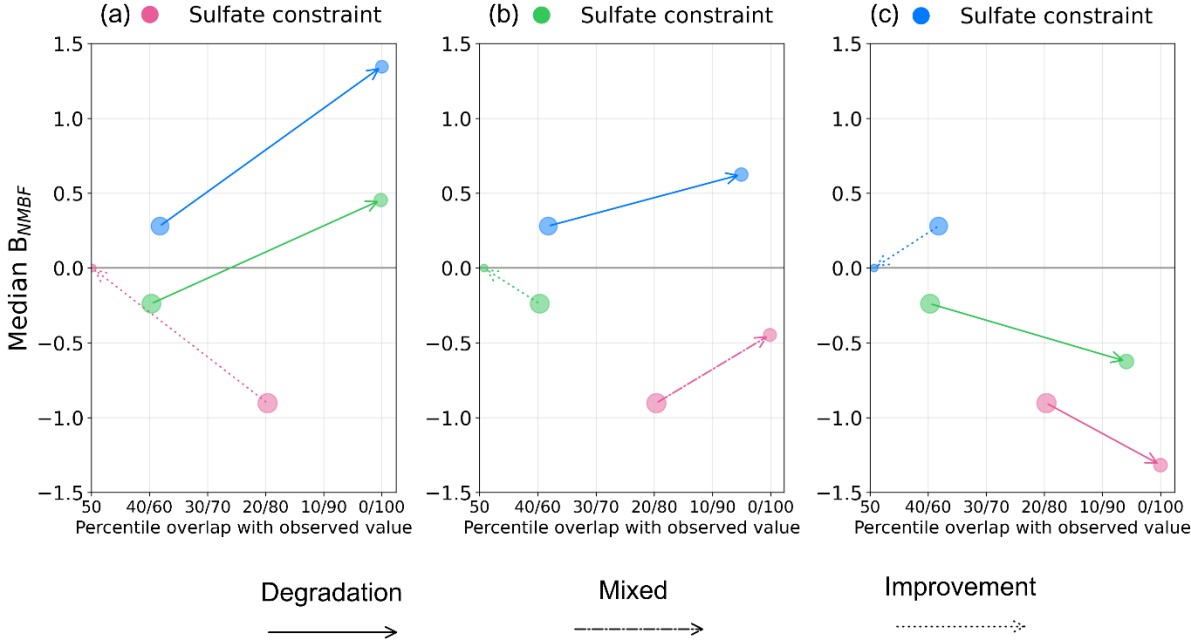

**Figure 9. Effect of regional sulfate observational constraints on model performance in sulfate clusters.** The x-axis represents the
percentile of the $B_{NMBF}$ distribution across model variants at which the observed value is located. The y-axis shows the median normalised
mean bias factor ($B_{NMBF}$) of the distribution. Panels represent the effect of (a) constraint to the pink cluster, (b) constraint to the green cluster,
and (c) constraint to the blue cluster, on all sulfate clusters. Arrows connect the positions of the unconstrained distribution (arrow start) to
the observationally constrained distribution (arrow end). Arrow line styles indicate the constraint's effect: solid for degradation in both
median $B_{NMBF}$ and percentile position of the observation, dotted for improvement in both, and dash-dot for improvement in one but
degradation in the other. For any distribution of model values with a positive median $B_{NMBF}$ (model > observation), the observed value
corresponds to a percentile less than 50 within that distribution (see Fig. 8).

We now examine which parameter values are ruled out by constraining sulfate in each cluster region (Fig. 10), to better

understand the inter-region inconsistency. After constraint, the pink and green clusters favour similar values across key

parameters except *cloud_drop_acidity* (Fig. 10a and b), suggesting this is the main parameter affecting sulfate differences

between them. In contrast, favoured parameter values differ more between constraints to the pink and blue clusters. For the

pink cluster (Fig. 10a), model variants that match high sulfate concentrations have lower cloud droplet acidity (promoting

sulfate formation from $SO_2$), lower dry deposition of sulfate and $SO_2$ (increasing aerosol lifetime and $SO_2$ concentrations), and

higher regional anthropogenic emissions (providing more $SO_2$ for conversion). In contrast, in Fig. 10c, model variants that

favour low sulfate concentrations in the blue cluster have higher cloud droplet acidity (suppressing sulfate formation from



SO$_2$) and mid-range dry deposition values, likely because sulfate is not strongly biased there, so lower deposition values are not more effective in bringing the model into agreement with observations.

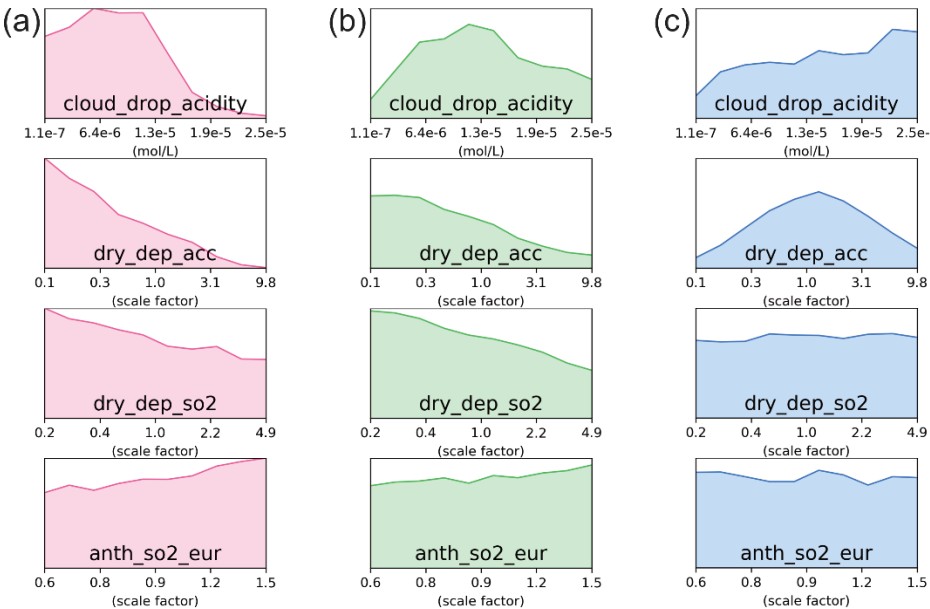


**Figure 10. Marginal probability density functions (PDFs) of model parameters after observational constraint**, for the 4 model parameters contributing most to sulfate concentration uncertainty across Europe (Fig. 4). PDFs are created using the input settings of the set of 5,000 model variants that best agree with observed sulfate concentrations in (a) the pink cluster, (b) the green cluster, and (c) the blue cluster. The y-axis scale is fixed for each parameter across panels to facilitate comparison between clusters: lower PDF values indicate a
greater reduction in model variants with those parameter values. Marginal probability density functions for all 37 parameters are shown in Fig. C1 (pink), C4 (green) and C2 (blue).

It is clear that our model is structurally incapable of representing regional variations in sulfate formation. Although the pink and green clusters favour similar parameter values after constraint (Fig. 10a and b), they remain inconsistent (pink and green arrows in Fig. 9a and b). In our simulations, *cloud_drop_acidity* is prescribed globally, so has no dependence on regional
atmospheric composition. Introducing a scheme that allows acidity to vary with composition (Turnock et al., 2019), would likely worsen the agreement: acidity would decrease in remote regions and increase sulfate production (blue cluster), while increasing in polluted regions and suppressing sulfate production (pink and green), contrary to the tendency required to match observations. Thus, *cloud_drop_acidity* alone cannot resolve the inconsistency; additional processes that vary regionally are needed to consistently match inter-cluster observations.






Another potential contributing factor to this inconsistency is that our PPE uses a simplified representation of $SO_2$ oxidation. The simulations include the gas-phase oxidation pathway (OH) and one aqueous-phase pathway ($O_3$), with their concentrations prescribed using monthly mean output from a fully coupled UKESM model run, averaged over the 1979–2014 period, and then perturbed in our PPE (*oxidants_oh* and *oxidants_o3*). Hydrogen peroxide, the dominant oxidant for aqueous phase $SO_2$

oxidation in winter (Gao et al., 2024), is only partially dynamic: its production and loss are modelled, but its concentration is limited by the prescribed oxidant fields and does not vary with regional conditions. Since hydrogen peroxide concentrations are typically higher in more polluted regions (green and pink clusters), $SO_2$ oxidation to sulfate in those areas may not be sufficient.

Altogether, the inconsistency highlights the need for interactive chemistry with appropriate regionally dependent oxidant production and sulfate formation mechanisms to provide the model with other alternatives to cloud droplet acidity for balancing regional sulfate concentrations. Using simplified, globally averaged chemistry is a common approach in many climate models to reduce aerosol complexity, but can lead to structural inconsistencies which limit the model's ability to match different regional observations at the same time. In the next section, we examine the consequences of this inter-regional inconsistency

in sulfate concentration.

### 3.5. Compromised constraint in the presence of structural inconsistencies

In this section, we explore whether a compromise between the two inconsistent constraints can be achieved by weakening the constraints applied to two seemingly inconsistent observations (Sect. 3.4). We achieve this by increasing the number of retained model variants. Retaining 5,000 model variants in our observational constraints is a subjective choice, which was designed to

reflect the presence of unquantified observational and emulator uncertainty (Sect. 2.6). This approach allows us to test whether the structural inconsistency persists under looser selection criteria, and to understand how the set of parameter values considered acceptable shifts when the constraints are relaxed.

Figure 11 shows the effect of retaining more model variants in a 2-d parameter space defined by the two most important (Fig.

4) and most strongly constrained (Fig. 10) parameters in the pink and blue clusters, *dry_dep_acc* and *cloud_drop_acidity*. Figure 11a shows that the pink and blue sulfate constraints are concentrated around opposite ends of the 2-d plane. While probability density functions overlap slightly along the diagonal, there are no model variants that satisfy both constraints at the same time. The overlap is an illusion caused by reducing the 37 parameter influences on sulfate concentration into a 2-d view





and indicates that a combination of the remaining 35 parameters is contributing to the structural inconsistency (so, visual
inconsistency would only be visible in higher dimensions). Relaxing the threshold to retain 25,000 (2.5 %, Fig. 11b) and 50,000
(5 %, Fig. 11c) of the original set of model variants weakens the constraints because the additional model variants retained
have larger biases on average. Retaining more model variants creates more visual overlap along the diagonal of the 2-d
marginal view in parameter space as the degree of compromise between constraints grows. However, no model variants satisfy
both constraints with this degree of compromise.


Agreement between the two constraints can only be achieved by more aggressively relaxing the two constraints to retain
235,000 model variants (23.5 %) in each case, a significant compromise. Even with this degree of compromise, only 422 model
variants (less than 0.2 % of those retained) satisfy both the pink and blue sulfate constraints (compromise shown in black in
Fig. 11d).


The combined inter-region sulfate constraint shows two main groups of parameter combinations, with a small third group
bridging the gap between them (Fig. 11d). Most model variants that fit the compromise have low *cloud_drop_acidity* and mid-
to high-range *dry_dep_acc*, whilst a smaller set have very high *cloud_drop_acidity* and very low *dry_dep_acc*. In the first
case, lower acidity allows more sulfate to form, while moderate to high dry deposition removes aerosol faster, leading to mid
to high sulfate concentrations. In the second group, higher acidity limits sulfate production, but very low deposition means
less is removed, resulting in mid to low concentrations. Both combinations produce similar sulfate levels through different
mechanisms; an example of equifinality, where multiple parameter combinations can produce the same model output (Beven,
2006). A small number of variants have mid-range values for both parameters, giving sulfate concentrations between the two
main groups. However, neither of these combined parameter effects corresponds to either of the individual cluster constraints
in Fig. 9. Marginal PDFs of all 37 parameters for this compromise are shown in Fig. C3.



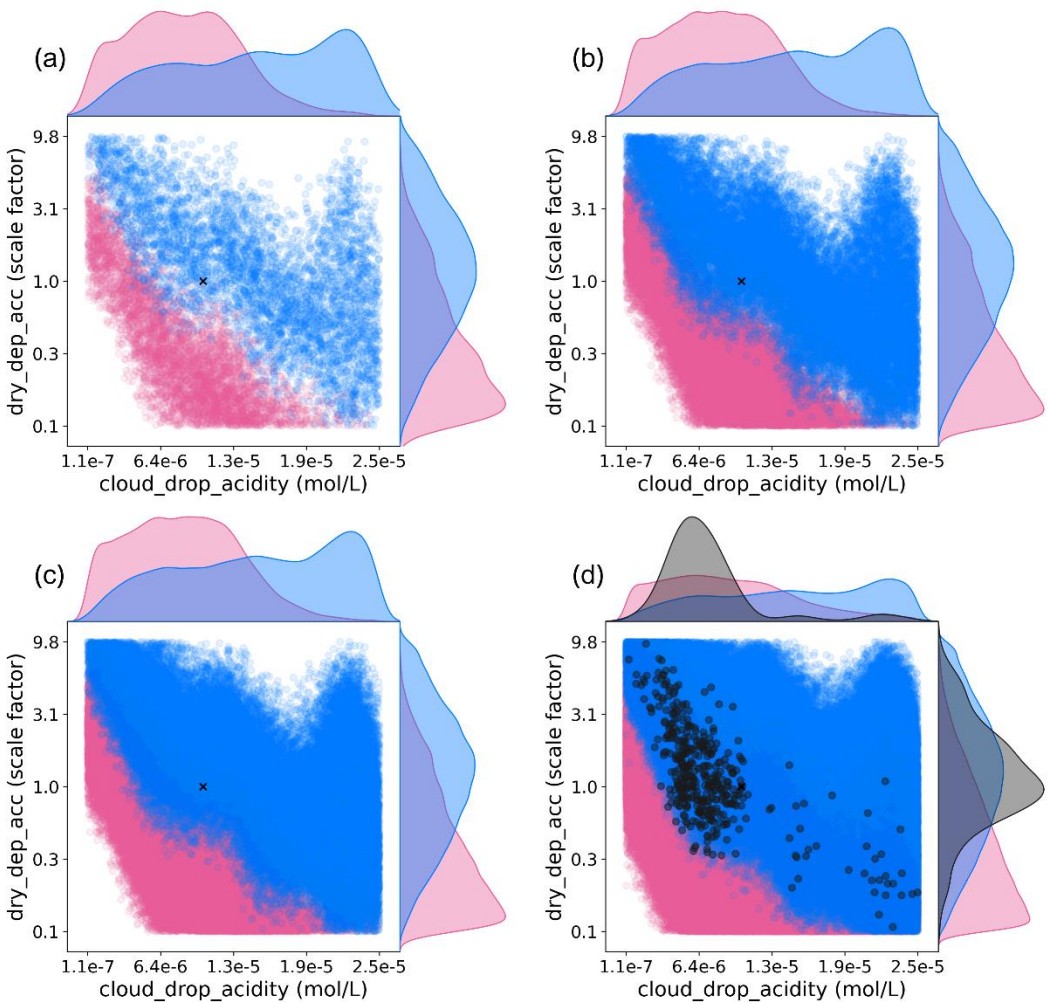

**Figure 11. Scatter plots indicating the density of constrained model variants over the marginal 2-d view of parameter space defined by the two parameters *dry_dep_acc* and *cloud_drop_acidity*.** Pink and blue points and associated distributions represent parameter values constrained to match sulfate observations in each cluster associated using the closest (a) 5,000, (b) 25,000, (c) 50,000 and (d) 235,000 model variants. The cross marks the combined parameter values used in the release version of UKESM1. In (d), the black points and associated distribution represent a subset of 422 out of 470 000 model variants that agree with observations in both pink and blue clusters.

Even when allowing a larger uncertainty in both constraints, the model is incapable of achieving a low bias in both regions at the same time. Figure D1a shows the effect of this compromise on all variables and clusters. On average, model variants are still negatively biased in the pink cluster (median $B_{NMBF}$ = -0.37), still positively biased in the blue cluster (median $B_{NMBF}$ = 0.21), and the observed values are outside the constrained distributions in both clusters. As a result, the compromise achieves



only tolerable agreement with observations in both clusters, rather than a close match in either. The constraint is therefore *no longer optimal*, consistent with the hypothesis proposed by Regayre *et al.* (2023) that structural inconsistencies demand a compromise in the tightness of constraint achieved. In addition, the compromise causes a strong degradation for most other

variables (Fig. D1). Except for $N_3$, where the initial overestimation is slightly reduced, the result is a reduced or null likelihood of matching observations, with observed values sometimes located outside the constrained distributions and an increased bias on average.

In summary, resolving the structural inconsistency between clusters requires considerably relaxing the strength of constraint

for each observation, thereby increasing the degree of model-to-observation error in each region. This process mirrors model tuning, or calibration, where agreement to multiple observations is balanced to mask the effects achieved via compromise (Elsaesser et al., 2025). Tuning and calibration approaches that neglect structural inconsistencies will achieve weaker overall constraints and a general degradation of model performance at simulating the state of the atmosphere. We suggest compromising model performance in this way is one of the key reasons climate projections of aerosol-cloud interaction forcing

have uncertainty that has persisted through several generations of climate model development.

### 3.6. Other potential structural inconsistencies

We categorise several other potential structural inconsistencies in this section, identified using constraints to match observations of the four variables in each cluster. Figure 12 and 13 extend the use of the constraint-effect metrics to show the effect of each constraint on all other clusters and variables (inter-cluster and inter-variable), as exemplified in Fig. 9 for sulfate

concentration inter-cluster constraints. Overall, there is very little consistency across the four variables over Europe. No variable within any cluster, when used as a constraint, reliably pushes other variables to better match observations within the same or other clusters. Sect. 3.6.1 and 3.6.2 cover the main inconsistencies.



**Figure 12. Effect of sulfate and SO₂ observational constraints on model performance (precision and accuracy) across all variables and clusters.** Marker colours correspond to regions in Fig. 4 and 5, with green for Central Europe, blue for Northern Europe/Scandinavia, and pink for UK/Spain. All other features are identical to Fig. 9.





**Figure 13. Effect of AOD and N₃ observational constraints on model performance across all variables and clusters.** All features are
identical to Fig. 12.




### 3.6.1. AOD-Sulfate inconsistency

Aerosol sulfate is a large component of AOD in polluted regions, so it is useful to evaluate their consistency. In our PPE, constraining either sulfate or AOD degrades model skill for the other variable. Sulfate constraints reduce model performance across all AOD clusters (Fig. 12a–c). In every case, constraint to sulfate reduces AOD agreement with observations. For the pink and green clusters, it also increases overall AOD bias (higher median $B_{NMBF}$), while for the blue cluster, it slightly reduces AOD median bias. Similarly, applying AOD constraints in any cluster reduces model skill across all sulfate clusters, and shifts distributions away from the observations (Fig. 13a–d). A similar issue was reported by Johnson *et al.* (2020), where joint constraints on AOD, PM$_{2.5}$, and sulfate led to conflicting parameter values and reduced the ability to constrain $\Delta F_{aer}$.

This inter-variable inconsistency is most clearly illustrated in the green AOD and sulfate clusters. On average, modelled AOD is overestimated and modelled sulfate is underestimated in the green clusters (Fig. 6). Therefore, constraint of AOD to match observations favours model variants associated with lower sulfate concentrations, which exacerbates the sulfate negative bias (Fig. 13b). In the other direction, constraint of sulfate to observations favours model variants associated with higher sulfate concentrations, which amplifies the existing positive AOD bias (Fig. 12b). However, unlike the inter-region inconsistency presented in Sect. 3.4, there exist combinations of the 37 parameters that match both sulfate and AOD observations at the same time without reducing the strength of constraint. Therefore, we classify this inter-variable inconsistency as level 1: the model can match the observations simultaneously, but the constraints do not converge, and the skill of the model is worse for both variables than if they were constrained separately (see Sect. 2.7).

Any adjustment to sulfate also affects AOD in our set of model variants. Therefore, there is limited flexibility to adjust AOD without affecting sulfate. Contributions to AOD in the model are dust, sulfate, sea salt, organic carbon, and black carbon aerosol, but only emissions of sea salt and sulfate, as well as dimethylsulfide aerosol precursor gasses (*dms*) were perturbed. AOD is also affected by the emission diameters of primary aerosol, which we perturbed.

Model variants that match higher observed sulfate concentrations in the green cluster are more likely to have relatively low values of both *dry_dep_so2* and *dry_dep_acc* (Fig. 14a). Lower *dry_dep_so2* allows more SO$_2$ to remain available for conversion to sulfate, while lower *dry_dep_acc* slows the removal of sulfate particles from the atmosphere. The *cloud_drop_acidity* parameter is also constrained towards central values to moderate aqueous-phase production of sulfate. In



contrast, model variants that match lower AOD are more likely to have low sea salt emissions (*sea_salt*) and lower sulfate

concentrations, achieved through higher *cloud_drop_acidity* and higher *dry_dep_acc* values (Fig. 14b).

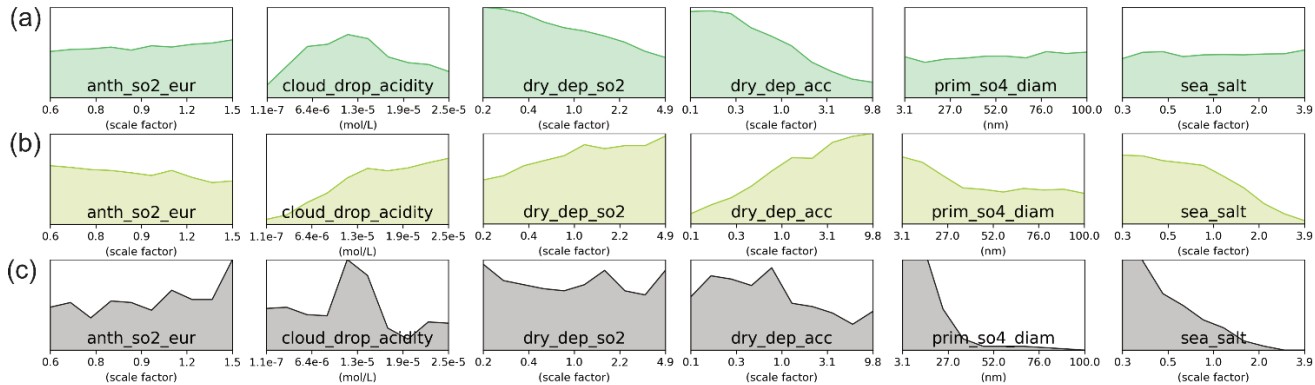

**Figure 14. Marginal PDFs of six key model parameters contributing to the AOD–sulfate inconsistency in the green cluster (Central**
**Europe)**. Panels show posterior PDFs after applying observational constraints for (a) AOD (5,000 variants), (b) sulfate concentrations (5.000
variants), and (c) the intersection of both constraints ("compromise"; 333 variants). The y-axis scale is fixed for each parameter across panels
to facilitate comparison between clusters: lower PDF values indicate a greater reduction in model variants with those parameter values. PDFs
for all 37 parameters are shown in Fig. C4 (sulfate), C5 (AOD), and C6 (compromise).

Constraints to AOD and sulfate in the green clusters lead to conflicting values for *dry_dep_acc* and *dry_dep_so2*. To increase
sulfate, both parameters are more likely to be low, whereas to reduce AOD they are more likely to be high. As a result, when

forcing a compromise between the inconsistent constraints (Fig. 14c), *dry_dep_acc* and *dry_dep_so2* remain effectively

unconstrained. Instead, *sea_salt* and *prim_so4_diam* are pushed towards extreme values as the only remaining degrees of

freedom for reducing AOD without further degrading sulfate concentrations. In particular, *prim_so4_diam* is constrained to

extremely low values deemed observationally implausible in previous work (Regayre et al., 2023). Figure D1b shows how this

compromise affects $N_3$, where median $B_{NMBF}$ increases sharply to 14 in the green cluster and 8 in the blue cluster

(approximately 3.5 times higher than before the compromise).

In UKESM1, GLOMAP does not account for ammonium nitrate emissions nor chemistry (Mann et al., 2010; Mulcahy et al.,

2020). Observational studies show that nitrate can account for a large fraction of $PM_{2.5}$ in Europe during winter (Ricciardelli

et al., 2017; Salameh et al., 2015) and that $PM_{2.5}$ correlates strongly with AOD (van Donkelaar et al., 2010). This omission

likely contributes to the AOD-Sulfate inconsistency by placing excessive burden on sulfate and sea salt to explain observed

AOD. A nitrate aerosol scheme is available in recent model versions (Jones et al., 2021), which may help address this

inconsistency. Furthermore, carbonaceous aerosol emissions were not perturbed in this PPE (Regayre et al., 2023). Expanding



the set of perturbed emissions in future PPEs will help determine whether the apparent inconsistency reflects incomplete
exploration of parameter space rather than a structural limitation.

### 3.6.2. SO₂ inter-cluster inconsistency

Existing structural limitations already lead to $SO_2$ overestimation in every cluster (Sect. 3.1; Mulcahy *et al.* 2020), and our
suggested change to reduce sulfate inter-region inconsistency (e.g. interactive chemistry; Sect. 3.4) could reduce biases
across all $SO_2$ clusters. Here, we use our PPE to uncover potential additional factors driving $SO_2$ inconsistency beyond these
known limitations.

$SO_2$ constraint effects suggest a structural inconsistency between clusters. On average, modelled $SO_2$ concentrations are
overestimated in all three clusters. However, constraining $SO_2$ towards the pink cluster reduces agreement with observed values
in the green and blue clusters (Fig. 12d). Similarly, constraining $SO_2$ in the green or blue clusters makes the bias in the pink
cluster worse and results in a less central percentile position for the observation (Fig. 12e and f). This is evidence of a level 2
inter-cluster inconsistency: no model variants can simultaneously match observations in individual clusters.

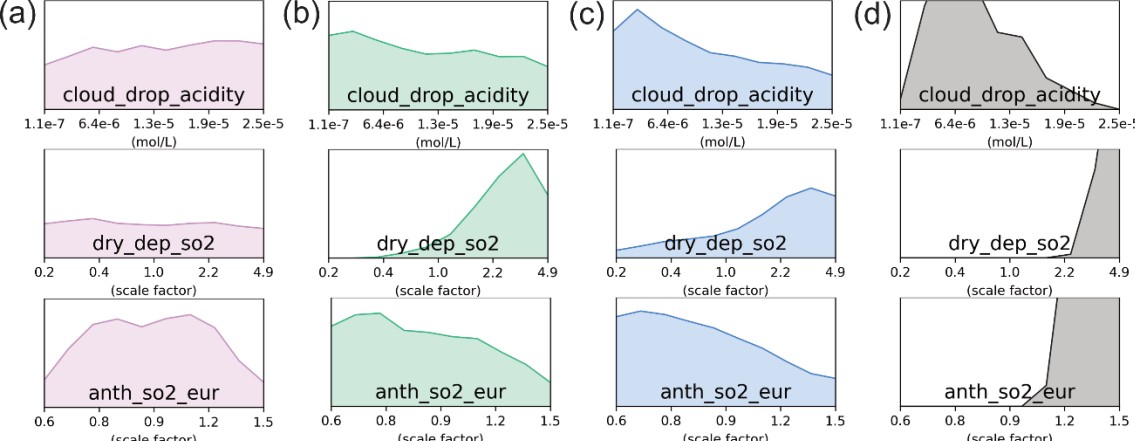

**Figure 15. Marginal PDFs of three model parameters contributing to SO₂ uncertainty after applying observational constraints.**
Shown for (a) the pink cluster (UK/Spain), (b) the green cluster (Central Europe), (c) the blue cluster (Northern Europe), each with 5,000
variants, and (d) their intersection ("compromise"; 386 variants). The y-axis scale is fixed for each parameter across panels to facilitate
comparison between clusters: lower PDF values indicate a greater reduction in model variants with those parameter values. PDFs for all 37
parameters are shown in Fig. C7 (pink), C8 (green), C9 (blue) and C10 (compromise).

Constraint to observed $SO_2$ in the pink cluster does not strongly rule out any part of the parameter space, as $SO_2$ concentrations
were already close to observations (Fig. 15a). Model variants that match observed $SO_2$ concentrations in the green and blue



clusters are more likely to have low *cloud_drop_acidity*, fast dry deposition of $SO_2$ (high *dry_dep_so2)*, and low anthropogenic $SO_2$ emissions from Europe (*anth_so2_eur*), all of which help reduce $SO_2$ by increasing removal or reducing emissions (Fig. 15b and c). When all three clusters are constrained together, the result is a combination of low *cloud_drop_acidity* and extreme high values of *dry_dep_so2* and *anth_so2_eur*, which are seemingly inconsistent with the individual constraints (Fig. 15d).

The compromise in parameter constraints can be understood by considering which parameters contribute to uncertainty in each cluster. The combined constraint favours model variants with high *dry_dep_so2*, which increases the removal rate of $SO_2$ from the atmosphere and reduces biases in the green and blue clusters. However, high dry deposition would excessively lower $SO_2$ in the pink cluster where concentrations are already centred on observations. Instead, model variants with very high *anth_so2_eur* are favoured to offset $SO_2$ removal in this cluster where anthropogenic emissions are relatively high. Increasing anthropogenic emissions to extreme levels has less effect on the green and blue clusters, where $SO_2$ concentrations are more sensitive to *dry_dep_so2* than to *anth_so2_eur* (Fig. 4b). As a result, model variants with extreme emissions and removal rates are retained in the compromise that matches observations in all three clusters, even though none of the individual constraints on their own would favour such extreme parameter values and are in fact more likely to have low *anth_so2_eur* values.

## 4.   Discussion and conclusions

This research is part of an overarching goal to develop a workflow for identifying opportunities for model development that address structural model deficiencies and enable more robust parametric uncertainty reduction in the UKESM1 aerosol scheme. Here, we (1) identified the main inconsistencies between aerosol observational constraints in European winter, (2) related these inconsistencies to likely structural deficiencies in the model, and (3) provided insight into the possible causes of these deficiencies.

We propose a generalisable workflow that uses inconsistencies between observational constraints as a diagnostic tool to identify underlying structural errors in the model. Our workflow (Fig. 16), identifies where combinations of constraints are inconsistent, and classifies their severity as either *level 2* or *level 1*. Level 2 inconsistencies (most severe), occur when no parameter combination can match observations across multiple aspects of the model. Level 1 inconsistencies (moderate) arise when some model variants can still match all observations, but where the constraints might improve skill in one aspect of the model, they lead to degradation in others.




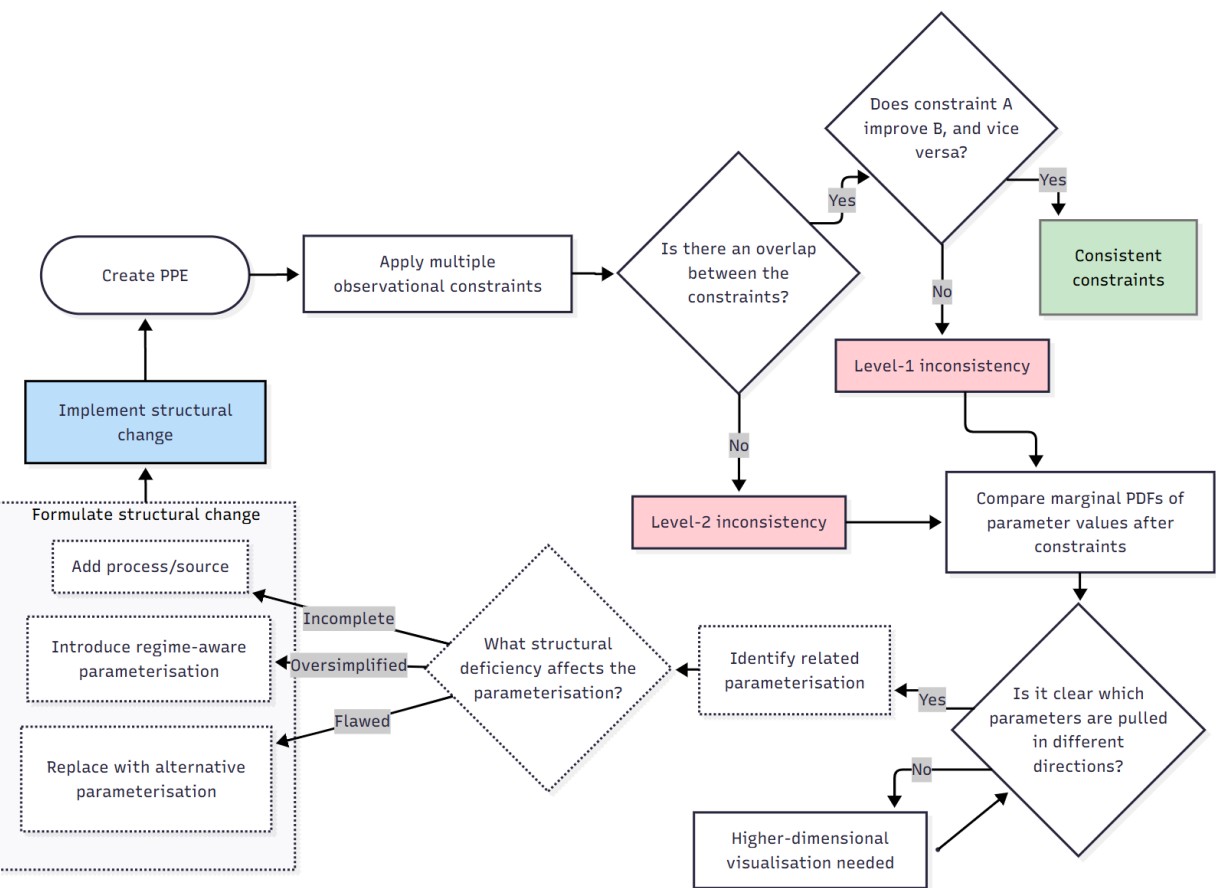

**Figure 16. Proposed workflow to identify structural inconsistencies between observational constraints, assess the structural deficiencies that cause them, and formulate corresponding structural model changes.**

Once inconsistencies are identified, we analyse which parameter combinations are retained by each individual constraint and their combination. Constraints that pull parameters in opposing directions provide actionable information on which parts of the model code are responsible for the inconsistency. This information allows us to trace the issue back to specific parameterisations and assess whether the way processes are represented is consistent with physical understanding. When constraints are in clear conflict, they often point directly to process-level assumptions that are missing, misrepresented, or oversimplified. However, not all inconsistencies lead to clear opposing trends in parameter space. The disagreement may be subtler, either because the constraint is weak or because it involves multiple interlinked parameters. Understanding these more complex cases would likely require higher-dimensional visualisation to identify how the inconsistency links back to multiple parameters and their interdependencies.






Identifying which parameterisations may be responsible for inconsistencies, and why, is not straightforward. In practice, this step relies on expert judgement supported by deep domain knowledge. It draws on a range of external sources, including laboratory studies, literature reviews and process-resolving simulations such as large eddy simulations (LES), to assess where existing parameterisations deviate from physical reality. These insights guide the formulation of informed structural change

hypotheses about ways to address the structural inconsistencies. Ideally, the process should be operationalised (Carslaw et al., 2025), with a cycle of proposed change implementation and repeat application of the inconsistency detection workflow. The intention is that each cycle will either reduce existing inconsistencies or reveal new ones that were previously hidden. Over time, this iterative process would gradually improve model structure and understanding.

To identify the main inconsistencies for our case study, we evaluated UKESM1 against aerosol observations by exploring the range of uncertainty in its input parameters. Observational constraints on sulfate, $SO_2$, AOD, and particle number concentration over Europe in winter revealed structural inconsistencies between regions and between variables in the same region. The main inconsistencies are summarised in Table 2. Most code changes suggested in Table 2 have already been implemented in newer model versions, so there is potential to test whether these structural changes improve consistency between observational

constraints in a new PPE. This next stage in the cycle of model evaluation and development would allow us to extend our analysis across seasons, regions, and additional variables to provide a more complete understanding of remaining model deficiencies related to inter-region, inter-variable and inter-seasonal inconsistencies.

**Table 2. Summary of key inconsistencies with corresponding level of severity, hypothesised structural deficiencies, and proposed**
**model developments to address them.**

| Inconsistency detected | Severity | Likely related structural deficiency | Potential solution |
|---|---|---|---|
| Sulfate inter-region | Level 2 | Lack of complexity in regional sulfate chemistry | Interactive chemistry (StratTrop, Archibald *et al.* 2020) |
| AOD-sulfate inter-variable | Level 1 | Missing aerosol emissions | Nitrate scheme (Jones et al., 2021), perturbation of carbonaceous aerosol emissions |
| $SO_2$ inter-region | Level 2 | Surface anthropogenic emissions; possibly compounded by Etna volcanic $SO_2$ treatment | Vertically distributed anthropogenic emissions, perturbation of *volc_so2* over lower values |





While we were able to identify potential structural inconsistencies where the model cannot simultaneously satisfy multiple observational constraints, we also encountered several overarching structural deficiencies where model values spanned by the PPE fail to match observations. For example, particle number concentrations were highly overestimated. $SO_2$ concentrations

were generally overestimated across Europe, likely because emissions were released at the surface rather than distributed vertically through the atmosphere (Ahsan et al., 2023). AOD was consistently overestimated in some regions, likely because of using mean volcanic $SO_2$ emissions that are much higher than occurred during the period we analysed.

We also tested the practice of forcing a compromise between inconsistent constraints. To do so, we relaxed individual

constraints until enough retained model variants could satisfy both sets of observations simultaneously. This approach was a way to explore what may happen when tuning structurally deficient models to match observations despite structural inconsistencies. We found clear limitations to this approach: it results in a model that performs only moderately well across most observed variables, risks "making the model right for the wrong reasons" (which can reduce skill in future climate simulations), and can worsen model performance for variables not included in the constraint process (e.g. particle number

concentration when constraining sulfate and AOD simultaneously). In this study, key inconsistencies persist even after increase the number of model variants kept in observational constraints, suggesting that the identified deficiencies cannot be explained by observational uncertainty alone. We consider this approach appropriate given that our aim is not to tightly constrain aerosol forcing, but to identify where the model fails to simultaneously match observations.

Our results add context to recent efforts to constrain $\Delta F_{aer}$ using PPEs and history-matching type techniques. Studies like Johnson *et al.* (2020) have shown that observational constraints can sometimes pull the model towards opposing values of aerosol forcing, limiting the reduction in uncertainty, whilst Regayre *et al.* (2020) showed observations have unequal value as model constraints. Regayre *et al.* (2023) identified a very small subset of observational constraints as suitable for narrowing aerosol forcing uncertainty, because many were inconsistent when applied together. The structural deficiencies and

inconsistencies we identified in this paper help explain why such limitations arise. By revealing where and why the model cannot simultaneously match observations, we suggest that this method reveals which aspects of the model need improvement to make more observational constraints usable in constraint-based efforts to reduce aerosol forcing uncertainty. Combined with efforts to apply observational constraints that align with the dominant causes of aerosol forcing uncertainty (Regayre et al., 2025), this approach should bring us closer to achieving the maximum feasible reduction in $\Delta F_{aer}$ (limited by observational

uncertainty, emulator uncertainty, and representation errors), and improve our ability to confidently constrain future climate change and inform policy decisions.

We suggest that progress in model development should prioritise the identification of structural deficiencies, rather than increasing model complexity (and associated uncertainty) without sufficient justification. The workflow presented in this paper

(Fig. 16) supports a shift toward a more evidence-based model development approach that prioritises changes most likely to reduce uncertainty and improve predictive skill.

Ultimately, this analysis framework would benefit from being extended across multiple models. For example, multi-model PPE efforts use several models with different structures while sampling similar sources of parametric uncertainty. Applying

the same observational constraints across these models would reveal differences in how consistently they match observations and allow more robust attribution of inconsistencies to structural deficiencies, based on the differences in process representations.

**Data availability**

The code used for this paper is available at: https://doi.org/10.5281/zenodo.17142337 (Prévost, 2025). Output from the A-

CURE PPE is available on the CEDA archive (Regayre *et al.*, 2022). Observational data used in this study were accessed from EBAS (https://ebas.nilu.no) hosted by NILU. Specifically, the use included data affiliated with the frameworks: ACTRIS, CREATE, EMEP, GAW-WDCA. The GHOST dataset is made freely available via the following repository: https://doi.org/10.5281/zenodo.10637449 (Bowdalo, 2024a; Bowdalo *et al.*, 2024b).

**Author contributions**

LP, KC and LR developed the original ideas and conceptualised the study. Code for emulation was based on input from JJ and LR. Code for GAM analyses was adapted by LP from source code provided by LR. LP created the statistical emulators and GAM variances at the grid box level, processed observations, created K-means clusters and applied observational constraints. KC and LR provided overarching guidance and helped interpreting results. JJ, DM and SM contributed through discussions. The original draft was written by LP and reviewed by all co-authors. The work was supervised by LR, KC, DM, and SM.



**Competing interests**

At least one of the (co-)authors is a member of the editorial board of Atmospheric Chemistry and Physics.

**Acknowledgements**

The PPE that informed this research was created using the ARCHER UK National Supercomputing Service under project allocation n02-NEP013406. This work used JASMIN, the UK's collaborative data analysis environment (https://www.jasmin.ac.uk).

**Financial support**

LP was supported by the NERC Panorama Doctoral Training Partnership with additional CASE sponsorship from the Met Office (grant no. NE/S007458/1, project reference 2886996). LR was supported by the Met Office Hadley Centre Climate Programme funded by DSIT. We acknowledge funding from NERC under grants A-CURE and Aerosol-MFR (NE/P013406/1 and NE/X013901/1).

**Appendix A**

**Table A1. Description of the 37 parameters perturbed, from Regayre *et al.*, (2023).**

| Parameter name | Min | Max | Default | Description | Perturbation type |
|---|---|---|---|---|---|
| a_ent_1_rp | 0 | 0.5 | 0.23 | Cloud top entrainment rate scale factor | Physical atmosphere |
| ai | 0 | 5e-2 | 2.57e-2 | Scaling coefficient for ice mass dependence on diameter | Physical atmosphere |
| ait_width | 1.2 | 1.8 | 1.59 | Modal width of Aitken modes | Aerosol process |
| anth_so2_asi | 0.6 | 1.5 | 1 | Anthropogenic $SO_2$ emission flux scale factor – Asia | Anthropogenic aerosol emission |
| anth_so2_chi | 0.6 | 1.5 | 1 | Anthropogenic $SO_2$ emission flux scale factor – China | Anthropogenic aerosol emission |



| anth_so2_eur | 0.6 | 1.5 | 1 | Anthropogenic $SO_2$ emission flux scale factor – Europe | Anthropogenic aerosol emission |
|---|---|---|---|---|---|
| anth_so2_nam | 0.6 | 1.5 | 1 | Anthropogenic $SO_2$ emission flux scale factor – North America | Anthropogenic aerosol emission |
| anth_so2_r | 0.6 | 1.5 | 1 | Anthropogenic $SO_2$ emission flux scale factor – Rest of the world | Anthropogenic aerosol emission |
| autoconv_exp_lwp | 2.15 | 3.31 | 2.47 | Exponent of liquid water path in autoconversion power law | Physical atmosphere |
| autoconv_exp_nd | -3 | -1 | -1.79 | Exponent of cloud droplet concentration ($N_d$) in autoconversion power law | Physical atmosphere |
| bc_ri | 0.2 | 0.8 | 0.565 | Imaginary part of the black carbon refractive index | Aerosol process |
| bl_nuc | 0.1 | 10 | 1 | Boundary layer nucleation rate scale factor | Aerosol process |
| bparam | -0.15 | -0.13 | -0.14 | Coefficient of the spectral shape parameter (β) for effective radius | Physical atmosphere |
| bvoc_soa | 0.32 | 3.68 | 1 | Biogenic monoterpene production rate of secondary organic aerosol scale factor | Natural aerosol emission |
| carb_bb_diam | 90 | 300 | 110 | Emission diameter of carbonaceous aerosol from biomass burning sources | Natural aerosol emission |
| carb_ff_diam | 30 | 90 | 60 | Emission diameter of carbonaceous aerosol from fossil fuel sources | Aerosol process |
| carb_res_diam | 90 | 500 | 150 | Emission diameter of carbonaceous aerosol from residential sources | Anthropogenic aerosol emission |
| cloud_drop_acidity | 1e-7 | 2.51e-5 | 1e-5 | Cloud droplet acidity | Aerosol process |
| cloud_ice_thresh | 0.1 | 0.5 | N/A | Threshold of cloud ice water fraction for scavenging | Aerosol process |
| conv_plume_scav | 0 | 0.5 | 0.5 | Scavenging efficiency (fraction of aerosol removed) of Aitken mode aerosol in convective clouds | Aerosol process |
| c_r_correl | 0 | 1 | 0.9 | Cloud and rain sub-grid horizontal spatial colocation | Physical atmosphere |
| dbsdtbs_turb_0 | 0 | 1e-3 | 1.5e-4 | Cloud erosion rate | Physical atmosphere |



| dms | 0.33 | 3 | 1 | Dimethyl-sulfide emission flux scale factor | Natural aerosol emission |
|---|---|---|---|---|---|
| dry_dep_acc | 0.1 | 10 | 1 | Dry deposition velocity of accumulation mode aerosol | Aerosol process |
| dry_dep_ait | 0.5 | 2 | 1 | Dry deposition velocity of Aitken mode aerosol | Aerosol process |
| dry_dep_so2 | 0.2 | 5 | 1 | Dry deposition velocity of $SO_2$ | Aerosol process |
| kappa_oc | 0.2 | 0.65 | 0.65 | Hygroscopicity parameter ($\kappa$) for organic aerosol – affects wet diameter and clear-sky radiative flux | Aerosol process |
| m_ci | 0 | 3 | 1 | Ice fall speed scale factor | Physical atmosphere |
| oxidants_o3 | 0.7 | 1.3 | 1 | Offline oxidant $O_3$ concentration scale factor | Aerosol process |
| oxidants_oh | 0.7 | 1.3 | 1 | Offline oxidant OH concentration scale factor | Aerosol process |
| prim_moc | 0.4 | 6 | 1 | Primary marine organic carbon emission flux scale factor | Natural aerosol emission |
| prim_so4_diam | 3 | 100 | 150 | Emission diameter of 50 % of new sub-grid sulfate particles; remaining 50 % emitted into coarse mode | Anthropogenic aerosol emission |
| rain_frac | 0.3 | 0.7 | 0.3 | Fraction of cloud-covered area where rain removes aerosol | Aerosol process |
| sea_salt | 0.25 | 4 | 1 | Sea salt emission flux scale factor | Natural aerosol emission |
| sig_w | 0.25 | 1.75 | 1 | Standard deviation of shallow-cloud updraft velocity scale factor | Aerosol process |
| two_d_fsd_factor | 1 | 2 | 1.4 | Scale factor for cloud condensate variance–cloud cover–convection relationship | Physical atmosphere |
| volc_so2 | 0.71 | 2.38 | 1 | Volcanic $SO_2$ emission flux scale factor | Natural aerosol emission |



# Appendix B

(a) Sulfate mass concentration

(b) SO$_2$ mass concentration

(c) Aerosol optical depth (440nm)

(d) Particle number concentration above 3nm

Emulator-to-Parametric Uncertainty Ratio

**Figure B1. Emulator uncertainty relative to the spread of emulated values across model variants.** The colour bar shows the ratio of the mean predicted standard deviation to the standard deviation of predicted means. Red grid boxes (metric > 1) indicate regions where emulator uncertainty exceeds the inter-variant spread and are excluded from further analysis. Blue grid boxes (metric ≤ 1) indicate more reliable emulator behaviour.







**Figure B2. Leave-one-out cross-validation (LOOCV) of emulator predictions for clusters defined in Fig. 4 and 5.** Each point represents the predicted (y-axis) versus the observed model output (x-axis), averaged over grid boxes with observations in the corresponding cluster. Error bars showing the emulator's predicted standard deviation. The dotted line indicates the 1:1 agreement line. LOOCV was performed by training the emulator while leaving out one of the 221 PPE members at a time, then predicting its output.



## Appendix C



**Figure C1. Marginal PDFs for all 37 parameters after constraint towards sulfate concentrations observations in the pink cluster (Western Europe).**

925





**Figure C2. Marginal PDFs for all 37 parameters after constraint towards sulfate concentrations observations in the blue cluster (Northern Europe).**





930

**Figure C3. Marginal PDFs for all 37 parameters after compromise between sulfate concentrations constraints in the pink and blue clusters.** The compromise consists of 422 common model variants after weakening constraint to individual clusters to 235,000 model variants each.







**Figure C4. Marginal PDFs for all 37 parameters after constraint towards sulfate concentrations observations in the green cluster (Central Europe).**







**Figure C5. Marginal PDFs for all 37 parameters after constraint towards AOD observations in the green cluster (Central Europe).**



**Figure C6. Marginal PDFs for all 37 parameters after compromise between AOD and sulfate concentration constraints in the green cluster.** The compromise consists of 333 common model variants after weakening constraint to individual clusters to 40,000 model variants each.





**Figure C7. Marginal PDFs for all 37 parameters after constraint towards SO₂ concentrations observations in the pink cluster (Western Europe).**







**Figure C8. Marginal PDFs for all 37 parameters after constraint towards SO₂ concentrations observations in the green cluster (Central Europe).**







Figure C9. Marginal PDFs for all 37 parameters after constraint towards SO₂ concentrations observations in the blue cluster (Northern Europe).



**Figure C10. Marginal PDFs for all 37 parameters after compromise between the pink, green and blue SO₂ concentration constraints.**
The compromise consists of 386 common model variants after weakening constraint to individual clusters to 55,000 model variants each.

955



## Appendix D

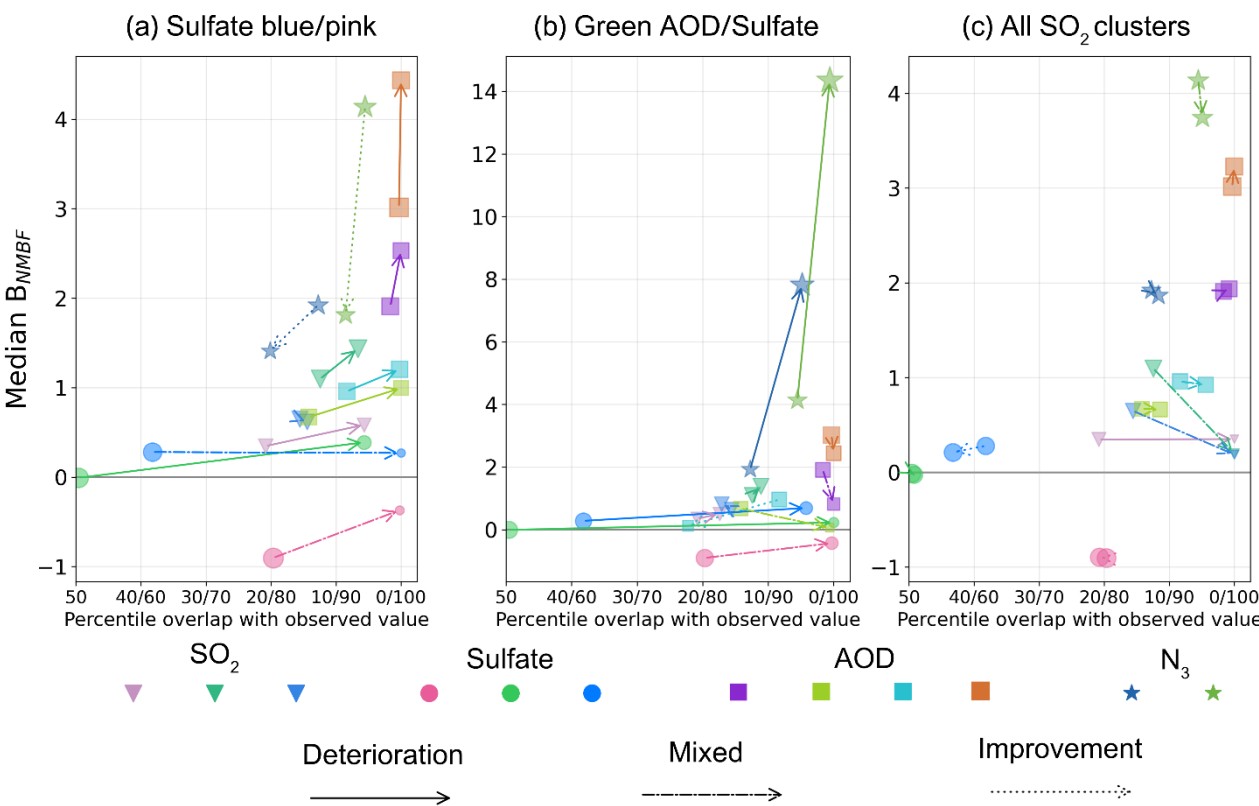

**Figure D1. Effect of compromise between observational constraints on other variables/clusters. (a)** Compromise between the pink and blue clusters for sulfate concentrations, with marginal PDFs shown in Fig. C3 (422 model variants). **(b)** Compromise between AOD and sulfate in the green cluster, with marginal PDFs shown in Fig. C6 (333 model variants). **(c)** Joint constraint for the pink, green and blue cluster for $SO_2$ concentrations, with marginal PDFs shown in Fig. C10 (386 model variants).

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
