# Peer review of "Detection of structural deficiencies in a global aerosol model to explain limits in parametric uncertainty reduction"

_EGUsphere, 2025_

## Author Comment (AC1)

We thank both referees for their thoughtful and constructive suggestions.

The referee reports are shown in **blue**, and our responses are shown in **black**. Changes to the manuscript are displayed using tracked changes: red underlined text indicates additions, and scored-out text indicates deletions.

During the review process, we identified an error in the wavelength used for AOD. We had mistakenly used 380 nm instead of 440 nm when comparing model output with observations. Correcting this mistake reduces the average AOD across Europe, which in turn reduces the degree of overestimation of AOD compared to observations. All figures and numbers cited in text have been updated with minor modifications. This correction does not affect the analysis or the conclusions of the study.

**Table 1:** Corrected median AOD $B_{NMBF}$.

| Variable | Pink Cluster, Western Europe | Green Cluster, Central Europe | Blue Cluster, Northern Europe | Orange Cluster, Southern Europe |
|---|---|---|---|---|
| **Sulfate** | -0.90 | -0.01 (-0.24) | 0.28 | – |
| **SO₂** | 0.35 (-0.02) | 1.10 (0.80) | 0.65 (0.43) | – |
| **AOD** | 1.36 (1.30) | 0.36 () | 0.64 | 2.29 (1.96) |
| **N₃** | – | 4.14 (2.66) | 1.92 | – |

In Figure 14b and c, the constraints on parameter values are less strong, as AOD is on average closer to observations. However, parameters are constrained towards the same part of the parameter space.

**Figure 14 (corrected):**

[Figure]

In Figure D1b, joint constraint to AOD and sulfate still leads a sharp increase in median $B_{NMBF}$ for $N_3$ clusters, although it is smaller than previously calculated (8, compared to 14). We have changed line 774 to reflect this change.

**Figure D1b (corrected):**

[Figure]

**Line 774:** Figure D1b shows how this compromise affects $N_3$, where median $B_{NMBF}$  sharply doubles in the green cluster and  the blue cluster.

Referee 1

The authors provide a workflow for identifying structural uncertainties. They use emulators to create surrogate models and constrain plausible parameter combinations. By examining inconsistencies in observational constraints across variables and regions, they trace these inconsistencies back to the underlying parameterisations, links them to likely structural model issues, and explores their possible causes.

The paper is well written and the figures provide sufficient visual context. There are a few minor concerns that the authors should address before publication.

Comments:

**Line 23:** region to regions

Changed.

**Line 25:** repeated **"them"** is vague — does it refer to inconsistencies or parameterisations?

We have revised the sentence to clarify that the repeated 'them' refers to structural deficiencies.

**Line 25:** By examining the parameter sets retained by constraints, we trace inconsistencies to the parameterisations that may cause them and propose targeted changes to address the underlying deficiency.

**Line 108:** suggested

Changed.

**Line 216.** Give a brief definition of Generalised Additive Models (GAMs)

We have added a sentence defining Generalised Additive Models.

**Line 218:** The importance of each parameter as a cause of model uncertainty was estimated using Generalised Additive Models (GAMs) . GAMs are flexible statistical models that represent the relationship between predictors and a response as a sum of smooth, linear or non-linear functions. We fitted non-linear GAMs to emulated model output for each variable within individual grid boxes using the *pygam* Python package (Servén and Brummitt, 2018). The fitted GAM functions were used to quantify the variance in model output attributable to each parameter, while allowing for non-linear effects (Strong et al., 2014), following Regayre et al., (2025).

**Line 238:** Does "six" refer to the number of grid boxes? Alternatively, is the question about how the number of clusters can be compared with the size of the region?

We clarified that 'six' refers to the number of clusters used when first applying k-means clustering.

**Line 242:** The number of clusters was chosen iteratively: we began with a high number relative to the size of the region (e.g. six clusters for Europe) and reduced it if clusters showed redundant patterns in dominant parameters and their contributions.

**Line 255:** Is linear interpolation applied spatially or temporally during collocation?

We do not use linear interpolation to calculate model-observations bias. The cluster-mean average only uses model values from grid boxes that match observation site locations only. However, grid boxes are compared to point locations from observations which contributes to representation error. Temporally, both in-situ observation and model output are monthly means. We have removed the word "collocated" in the text and clarified this paragraph to avoid any misunderstanding.

**Line 256:** Model–observation bias is calculated for each model variant ($i$ = 1 to 1,000,000) using normalised mean bias factors following Yu *et al.*, (2006). N denotes the number of observational sites in the cluster. For each site j, we use a single observed value ($O_j$) and pair it with the modelled value ($M_{ij}$) from the grid box containing that site for every model variant i. Both observations and model values are monthly averages.

Here, N is the number of observational sites in the cluster. Each site contributes a single observed value O, collocated with one modelled value $M_i$ from each model variant. Thus, for a given model variant *i*, the cluster-mean model value is $\overline{M_i} = \frac{1}{N}\sum_{j=1}^{N} M_{ij}$ and the cluster-mean observation is $\overline{O} = \frac{1}{N}\sum_{j=1}^{N} O_j$. The normalised mean bias factor (B$_{NMBF}$) is then calculated as follows: [...]

**Line 275:** what is the definition of 'model variants that are common'?

We apply joint constraints by selecting the intersection between the two sets of model variants kept for individual constraints. We have clarified this definition in the text.

**Line 281:** For joint observational constraints, we identify the set of model variants that are common to all individual constraints that form the joint constraint. the model variants that are common to each individual constraint set.

**Line 299.** If the parameter space does not converge across different observational constraints, does this indicate structural uncertainty and suggest that the model structure needs refinement rather than relying on tuning?

This interpretation is correct. We have modified the text to make this point clearer.

**Line 305:** In the ideal case, all observational constraints would guide the model toward the same part of parameter space. That is, each constraint would support convergence towards parameter combinations that produce simulations consistent with several observed variables toward the parameter combination that best represents the real system. When constraints do not converge, it indicates that the model would need to be tuned differently to match each variable and that, having exhausted the parameter space, no model variant exists that is consistent with multiple observations. In history-matching terminology, this situation is referred to as the "terminal case" (Salter et al., 2019). Such lack of convergence suggests a structural deficiency rather than a problem that can be resolved through tuning alone. We therefore define this lack of convergence between constraints as a *potential structural inconsistency*. In such cases, the model is not realistic which suggests a potential structural deficiency. We define this lack of convergence between constraints as a *structural inconsistency*.

**Line 326-7:** Can the inconsistency also be attributed to emulator uncertainty?

Emulator uncertainty could contribute to apparent inconsistencies, particularly where emulators do not validate well or have large uncertainty. To account for these possibilities in this study, we validate the skill of our emulators and quantify their uncertainty across the parameter space as described in Appendix B. Additionally, we remove the small subset of grid boxes with high emulator uncertainty from the analysis. We also keep emulator skill validation in mind when interpreting results. However, we consider emulator uncertainty less likely to affect conclusions than the other uncertainties mentioned. We have updated the text to acknowledge it alongside other possible explanations.

**Line 336:** We interpret the existence of an inconsistency as evidence of a potential structural deficiency in the model. However, such an inconsistency is not definitive proof of structural error; other explanations are possible, including larger-­than-estimated observational error, or the possibility that important parameters have not been perturbed, or emulator uncertainty, especially for variables with lower emulation skill.

**Line 370 (Figure 3):** Does each circle or triangle represent one observational site collocated with a model grid box? You may want to clarify this in the caption.

Thank you for the suggestion.

**Figure 3 caption:** Observed values and their position within the PPE range in January 2017 across Europe for the four variables. Markers are located at observational sites, and each site is compared to the nearest model grid box. Triangles indicate observations outside the PPE range. Circles represent observations within the PPE range.

**Figure3:** The number of available data points in 3d ($N_3$) is significantly lower than in the other plots (sulfate, $SO_2$, and AOD). Could you clarify the reason for this difference? In addition, the spatial locations of data points do not appear to match across the observed variables. This brings me back to a previous question: are the observational data points collocated with the model grid when comparing observations to model outputs?

The lower number of data points for $N_3$ reflects the limited availability of observational data for particle number concentration in Europe during January 2017.

Regarding spatial locations, the observational networks for these variables are not identical, so site locations vary between datasets. For model–observation comparison, we use the model value at the grid box corresponding to each observational site (nearest grid box), without interpolation or averaging.

**Line 503-505:** Could the possible explanations also include removal processes (e.g., dry_dep_acc, cloud_drop_acidity) as indicated by Figure 5c?

We agree that underestimated removal processes could also contribute to the positive AOD bias. We have added this as an additional possible explanation in the text.

**Line 519:** The region-wide positive bias suggests that the model is systematically overestimating aerosol sources, size or radiative properties,

**Line 655-665:** you may want to label the three groups in Figure 11d. In Figure 11d, you could use boxes to highlight the different groups, so that it's easier to link the figure with the corresponding text descriptions.

Thank you for the suggestion. We have added two boxes to highlight the groups of different behaviour in the figure and modified the text accordingly.

[Figure]

**Line 675:** The combined inter-region sulfate constraint shows two main groups of parameter combinations,  (black

points; Fig. 11d). Most model variants that fit the compromise have low *cloud_drop_acidity* and mid- to high-range *dry_dep_acc* (box 1; Fig. 11d), whilst a smaller set have  high *cloud_drop_acidity* and  mid- to low-range  *dry_dep_acc* (box 2; Fig. 11d).

**Figure 2B:** The emulator uncertainty for $N_3$ seems large. Does this substantially affect your observational constraints?

The emulator uncertainty for $N_3$ (Fig. B2) is indeed larger, primarily because the emulator tends to underpredict very high simulated $N_3$ values (>30,000 cm$^{-3}$). However, this does not substantially affect the observational constraint based on $N_3$ because the observations themselves are much lower than these extremes (<3500 cm$^{-3}$). Constraining to observations selects model variants with $N_3$ values well below 10,000 cm$^{-3}$, which the emulator predicts accurately.

The underprediction of high $N_3$ does influence the figures: the median $B_{NMBF}$ for $N_3$ in Figures 6, 12, 13 and D1 may in reality be higher than shown. This is acceptable because $N_3$ constraints are not central to our analysis. $N_3$ is included primarily to illustrate the negative consequences of compromising between structurally inconsistent constraints (Section 3.6.1), where joint constraint to AOD and sulfate leads to low values of *prim_so4_diam*, causing $N_3$ to increase sharply. In line 774, we show that the joint AOD–sulfate constraint selects model variants with high $N_3$, which results in the median $B_{NMBF}$ for $N_3$ doubling. However, these model variants correspond to the region where the emulator underpredicts $N_3$. With a perfect emulator, the increase in median BNMBF would likely be even greater.

We have added the following statements to clarify the consequences of emulator underprediction.

**Line 773:** Figure D1b shows how this compromise affects $N_3$, where median $B_{NMBF}$ sharply doubles in the green cluster in the blue cluster. However, high $N_3$ values tend to be underpredicted by the emulator (Fig. B2), so the true increase in median $B_{NMBF}$ would likely be even greater.

Referee 2

Review of "Detection of structural deficiencies in a global aerosol model to explain limits in parametric uncertainty reduction" by Prévost et al.

This study describes a framework for diagnosing and identifying potential sources of structural uncertainties within models using a combination of Perturbed Parameter Ensemble (PPE) data (generated using the UKESM) and observational data constraint. The structural uncertainty in question focuses on aerosol-radiation parameterizations

within UKESM, and targets European winter observations of sulfate aerosol, sulfur dioxide, AOD, and aerosol particle number concentration. The framework outlined here (1) carefully identifies the key causes of parametric uncertainty in the simulation of the four observable quantities mentioned above; this uncertainty then (2) informs k-means spatial clustering of key parameter influence in three to four main modes over Europe which are then emulated to produce a robust array of parameter combinations for each cluster and calculate emulator bias in corresponding observational datapoints; next (3) observational constrains are applied to corresponding clusters to isolate members that reduce bias while simultaneously identifying how the single cluster constraints contribute to other cluster bias and illuminate structural bias through diagnosis of model precision vs accuracy; finally, this leads to (4) exploration of structural uncertainties and their potential causes.

This work had several key findings that stood out. One is that constraint to observations in a given sector does not necessarily improve model-observational agreement in other sectors. While this is not the first study to describe this, they support previous work with similar findings (summarized in their introduction) by adding a quantitative analysis of this phenomenon. This also informed inter-sector comparisons showing that the constraints within a sector could lead to competing behavior in different variables. In some cases, improving overall model constraints meant greatly weakening the constraint to observations, introducing more error in the model-to-observation comparisons. This all laid the groundwork for characterizing structural uncertainty in aerosol-radiation interactions in UKESM based on cluster and parameter overlap, identifying different degrees of structural error and potential causes.

While my expertise doesn't lie in many of the statistical methods applied in this work, I found the paper compelling, interesting, well written, and logically organized. I thought the findings were well-supported by analysis and references, and the figures and application told a cohesive story. The one concern I had stems from a question that I became a bit fixated on as I read the paper: how do you know if the uncertainty is structural or related to the design of your PPE (i.e., missing parameter(s))? In some regards, this is still a structural uncertainty, but it comes from the structure of your PPE instead of your model. The authors do acknowledge the gray area of the structural uncertainty quantification before diving into their results as well as identifying parametric choice as a possible structural flag (they mention this briefly in Section 3.6.1). However, I would have liked to hear their thoughts regarding the potential contribution to inferred structural uncertainty that could come from the actual PPE design. Perhaps this was deemed small through history matching? This was unclear to me, and barring some misunderstanding on my part, it seemed an especially important part of the discussion to include in Section 3.6.1 and in the flowchart. Please see my one major comment and the minor comments below for more information.

Overall, I recommend this paper be accepted after minor revisions.

Major comment:

It seems like the PPE design can have strong implications for the structural uncertainty quantification proposed herein, and the degree to which this is contributing wasn't always clear. This seemed most apparent when reading the interpretation of the AOD-Sulfate discrepancy in Section 3.6.1, where the authors do mention that this spread could be caused by lack of exploration of the parameter space. While they mention that nitrate and carbonaceous emissions may be factors, could this also be due to a lack of dust emission and RI parameters in their PPE? Assuming dust has a significant contribution to AOD in the European winter along with nitrate and carbonaceous aerosol, if their PPE had included dust emission and RI parameters it seems the discrepancy between AOD and sulfate may not have existed as dust could have been changed within its uncertainty while sulfate could remain unchanged (same for nitrate and carbon).

I think this should be explored briefly by the authors through some supplemental description of aerosol species contributions to AOD across their time period. If dust or carbonaceous aerosol have a larger impact than sulfate then this may inform the PPE design.

I would also appreciate a bit more discussion as to the role that the PPE plays in interpretation of structural error. If it is significant and could be identified by some key characteristics such as the divergent parametric behavior noted in Section 3.6.1, it might also be worth adding a connection in Fig. 16 (potentially between "Identify related parameterization" and "create PPE") that describes some expert elicitation on PPE design. Please see minor comments for in-text details.

We thank the reviewer for this comprehensive comment. The points raised here are expanded upon in the minor comments, where we address them in detail.

Minor comments

**Line 75:** Does history matching operate on an unchanging set of selected parameters that vary in their values (i.e., a single PPE), or does it operate on multiple PPEs with different parameter lists? Please clarify here and/or in the paper. This gets at a concern I have throughout reading this paper which is how one separates the unexplored parametric uncertainty (i.e., from missing parameters in your PPE) from structural uncertainty. Perhaps history matching gives some confidence in that separation, but if the list of parameters remains the same, it seems one may be missing or

mischaracterizing the parametric uncertainty from parameters that haven't been included in the parameter set.

History matching was not used in this study; it was mentioned in the introduction to provide context for alternative approaches to reducing parametric uncertainty. History matching typically operates on a single PPE with a fixed set of parameters. It does not generally involve multiple PPEs with different parameter lists.

History matching does not resolve the challenge of separating unexplored parametric uncertainty from structural deficiencies. The implausibility metric used in history matching accounts for uncertainty in observations, emulator uncertainty and structural uncertainty. The uncertainty arising from an incomplete PPE would need to be absorbed into the structural uncertainty term, which is very difficult to estimate and therefore often arbitrary.

We address the implications of incomplete PPE design in response to related following comments.

**Lines 282-284:** "it means that no amount of parameter retuning will bring the model into agreement with the observations" – this is true, but this is tied to the *chosen* parameter ranges. How much confidence is there in the preexisting parameter ranges, and could they be expanded? Also, could it be that the inclusion of another parameter to the PPE might change model sensitivity to preexisting parameter ranges, potentially changing the status of the structurally deficient members?

It is true that this statement only applies to the PPE design (chosen parameters and their ranges). However, our choice of parameters is based on more than a decade of evaluating causes of aerosol forcing uncertainty in several generations of PPEs with the same model. We aim to include a representation of all key processes that affect the responses analysed in our PPE design. Parameter responses and ranges are tested using one-at-a-time (OAT) tests, and parameters that were previously included but found to be inactive have been interchanged in and out of successive PPE designs. Furthermore, the elicited parameter ranges are deliberately wide to ensure that all realistic options are spanned.

**Line 289:** An observation outside the PPE range is a clear indication of the presence of a structural model deficiency, as it means that no amount of parameter retuning will bring the model into agreement with the observations, given the parameters that were included in the PPE and the wide range of values they were perturbed over.

**Lines 498-500:** Are you able to speak to the variation in AOD bias across Europe? Is the lower bias in Northern Europe related to being close to the source with more consistent seasalt exposure, while the other regions are impacted more by the more unique dynamical conditions that might transport seasalt into the mainland of Europe?

We agree that there is clear variation in AOD bias across Europe, with Southern and Western Europe having the largest positive bias and Northern and Central Europe having smaller bias. As noted in the text, the high bias in Southern Europe is likely influenced by overestimated volcanic $SO_2$ emissions (Fig. 3 and Sect. 2.6). It is true that both Northern and Central Europe clusters have sea salt as a dominant source of aerosol (new Fig. E1, explained below) and dominant contributor to parametric uncertainty (Fig. 4). However, it is difficult to give definite reasons for this variation in bias as we cannot quantify the extent of overestimation coming from excessive volcanic $SO_2$ emissions. The parameter range used for *volc_so2* was too narrow to capture the full uncertainty, hence limited our ability to assess its contribution to the bias. We plan to use a wider range in future PPEs, as outlined in Table 2.

**Line 522:** change 'parametrisations' to 'parameterisations'

Changed.

**Line 594:** "...model variants that match high sulfate concentrations..." - I'm fairly certain this is in reference to Figure 7, but it would be nice for the reader to have a reference to that figure for clarification.

Agreed. Added a reference to Figure 7.

**Line 610:** For the pink cluster (Fig. 10a), model variants that match higher than average sulfate concentrations (Fig. 7a) have lower cloud droplet acidity (promoting sulfate formation from $SO_2$), lower dry deposition of sulfate and $SO_2$ (increasing aerosol lifetime and $SO_2$ concentrations), and higher regional anthropogenic emissions (providing more $SO_2$ for conversion).

**Line 598:** "...likely because sulfate is not strongly biased there..." - Please justify this statement with a figure reference or clarification. Fairly certain you are referencing Fig. 2 but being more explicit in this section will make it easier to follow for the reader.

The statement refers to Figure 7, which shows that sulfate bias in the blue cluster is, on average, less pronounced compared to the pink cluster. We have revised the sentence to explain this point more clearly and added an explicit reference to Fig. 7.

**Line 613:** In contrast, in Fig. 10c, model variants with low sulfate concentrations in the blue cluster have higher cloud droplet acidity (suppressing sulfate formation from $SO_2$) and mid-range dry deposition values The constraints on parameter values for the blue cluster are weaker because bias there is, on average, smaller than in the pink cluster (Fig. 7c and a).

**Line 657:** Please consider referencing the black points in this citation. Something like: '(black points; Fig. 11d)'.

Agreed and added. Thank you for the suggestion.

**Line 708:** 'Aerosol sulfate is a large component of AOD in polluted regions...' - Please quantify in supplementary or cite where this statement comes from. How large a contribution does sulfate have on average to AOD in the time periods analyzed here? I think something like a mass weighted contribution of all aerosol species to AOD could serve as a good reference.

Thank you for the suggestion. We have added Fig. E1 in the appendix, which shows the contribution of each modal aerosol species to total mass mixing ratio. In the green cluster, which this statement referred to, the main aerosol components are sea salt (ranging from 50% to 70 %), sulfate (ranging from 20 to 45%) and organic carbon (ranging from 15% to 30%). We have revised the sentence to indicate that sulfate is an important component of AOD in this cluster rather than the dominant one, and we now reference Fig. E1 to support this statement.

**Line 730:** Sulfate aerosol is an important component of AOD in Central Europe (contributes from 20 to 45% of total aerosol mass mixing ratio, Fig. E1),  so it is useful to evaluate their consistency.

**Additional figure:**

[Figure]

**Figure E1. Percent contribution of individual aerosol species to the total column aerosol mass (kg m⁻²) in January 2017 across Europe, based on the PPE ensemble mean.** Maps show the spatial distribution of contributions from (a) sea salt, (b) sulfate, (c) black carbon, and (d) organic carbon. Percentages are calculated from vertically integrated modal aerosol mass mixing ratios. Dust is not included because it is not represented as a modal species in the model.

**Line 726-729:** This is interesting. I wonder if a high sensitivity to dust is driving your AOD overestimation. This may be structural, but it seems it could also be parametric. On this note, how do you differentiate structural deficiencies from parametric uncertainty that wasn't addressed? Couldn't a dust emission parameter be contributing to structural uncertainty due to its not being included in the perturbed parameter list?

Thank you for raising this point. We have added Appendix Fig. E2 to show the contribution of dust to total AOD. While dust can be an important component of aerosol in other seasons and regions, its contribution in European winter is minimal in our model (<5%), so it is unlikely to drive the AOD overestimation and to contribute to the inconsistency between AOD and sulfate.

**Additional figure:**

[Figure]

**Figure E2. Percent contribution of dust to total AOD in January 2017 across Europe, based on the PPE ensemble mean.**

We agree that dust emission flux could be an important parameter to include in future PPEs to provide a more complete analysis of aerosol uncertainty.

We have added a reference to Fig. E1 and Fig. E2 in this paragraph.

**Line 751:** Contributions to AOD in the model are  sulfate, sea salt, organic carbon,  black carbon aerosol (Fig. E1) and dust (Fig. E2), but only emissions of sea salt and sulfate, as well as dimethylsulfide aerosol precursor gasses (*dms*) were perturbed. AOD is also affected by the emission diameters of primary aerosol, which we perturbed.

**Line 758-760:** I think this statement is very important and may require additional elaboration. What stands out to me is that dust/nitrate/carbonaceous emissions and dust optical properties were not perturbed within your PPE framework, both of which could have a large impact on your AOD. I'm not sure how sensitive dust in the UKESM is to meteorological conditions or if it is directly emitted, but I see this as a potential target in this comparison. If dust is indeed the culprit for the AOD overestimate, then sulfate is getting pushed into unrealistic concentrations to account for it. This is also a parametric source of uncertainty, but does this get lumped in with structural uncertainty by virtue of its not being included in the parameter list?

As noted above, dust emissions and optical properties were not perturbed in this PPE, but we have shown that dust contributes only minimally to AOD in this study. Nitrate was not included in the model and therefore could not be perturbed, and is indeed one of the potential structural deficiencies that we later identify. We agree that the omission

of carbonaceous emissions is significant, particularly given the large contribution of organic carbon to AOD in the green cluster (Appendix Fig. E1). This omission could contribute to the inconsistency described in Section 3.6.1, as discussed in Line 784. We have elaborated the statement in the manuscript.

**Line 785:** Furthermore, carbonaceous aerosol emissions were not perturbed in this PPE (Regayre et al., 2023) , even though organic carbon accounts for approximately 15–30% of total aerosol mass mixing ratio in this cluster (Fig. E1). Including carbonaceous emissions in future PPEs will be important for assessing whether the apparent inconsistency reflects incomplete exploration of parameter space rather than a structural limitation.

We have also added a paragraph in the Discussion section acknowledging that the presence of unexplored parametric uncertainties contributes to ambiguity when diagnosing structural deficiencies with the current workflow.

**Added paragraph, line 852:** The effectiveness of our workflow as a tool for detecting structural deficiencies depends on the completeness of the PPE design. If important sources of parametric uncertainty are omitted, inconsistencies can fall into a grey area where it is difficult to tell whether they arise from unexplored parameter space or from genuine structural error. Future PPEs should aim for more comprehensive coverage of uncertain processes to reduce this ambiguity and improve confidence in attributing inconsistencies to structural deficiencies.

**Line 861:** Ideally, the process should be operationalised (Carslaw et al., 2025) as an iterative cycle: applying the inconsistency detection workflow, implementing structural changes and refining parameter space coverage, then reapplying the workflow to evaluate improvements. The intention is that each cycle will either reduce existing inconsistencies or reveal new ones that were previously hidden.  This iterative process should gradually improve model structure and understanding over time.

**Figure 16,** 'identify related parameterization' ◊ 'what structural deficiency affects the parameterisation': Could another option be that the parameter space may be missing a key contributor to the chosen model diagnostic (i.e., AOD)? In this case, would it be necessary to reformulate the PPE or add an additional parameter? Not sure how to identify when this is the case as it may be considered a structural issue but I see this as a potential addition to this workflow.

Thank you for the useful suggestion. We have added a node in the schematic to suggest that refining the coverage of sources parametric uncertainty during the PPE design is an important step in this iterative workflow.

[Figure]

**Line 815:** "...they often point directly to process-level assumptions that are missing, misrepresented, or oversimplified." - I think this is a key comment that could be built on for interrogating the PPE design.

The statement was intended to describe how inconsistencies between constraints can be used to diagnose structural deficiencies, such as missing or oversimplified process representations, assuming the PPE design includes all relevant sources of parametric uncertainty. We have not changed this section, as the limitation regarding incomplete PPE design is already addressed in the added paragraph on line 852 following a previous comment.